# SCOPE: Evolving Symbolic World for Planning in Open-Ended Environments

Yundaichuan Zhan [* 1]   Minghe Gao [* 1]   Zhongqi Yue [1]   Wendong Bu [1]   Wenqiao Zhang [1]   Guoming Wang [1]
Jisheng Dang [2]   Juncheng Li [1]   Siliang Tang [1]   Yueting Zhuang [1]

## Abstract

Recent works have explored integrating Vision-Language Models (VLMs) with classical planners that rely on symbolic representations of planning problem to generate long-horizon plans for complex embodied tasks. However, in open-ended environments, these symbolic representations obtained from perception are often incomplete, leading to suboptimal performance. To address this, we introduce SCOPE, a self-adaptive symbolic planning framework that supports refining action plans and evolving the symbolic world—the symbolic representations of open-ended environments. SCOPE comprises two synergistic modules: a Symbolic Execution Simulator (SESim) that conducts symbolic validation and real execution of action plans, leveraging the feedback to refine the plans and evolve the symbolic world; and a Self-Adaptive Symbolic Memory (SASMem) that further distills feedback into evolving symbolic knowledge to enhance long-horizon planning and modeling of the symbolic world. Experiments in open-ended environments show that SCOPE significantly improves the completeness of the symbolic world, the success rate of plans under environment perturbations, and cross-task grounding and adaptability across diverse embodied scenarios.

## 1. Introduction

Vision-Language Models (VLMs), powered by large-scale multimodal pretraining (Du et al., 2022; Driess et al., 2023; Alayrac et al., 2022; Achiam et al., 2023; Hurst et al., 2024; Yang et al., 2025), excel at locating objects (Chen et al., 2024; Ao et al., 2025; Zeng et al., 2022), recognizing spatial relations (Lu et al., 2016; Cheng et al., 2024; Fei et al.,

2024), and interpreting goal-relevant cues from images and text (Yang et al., 2023). These capabilities make VLMs strong candidates for serving as the agent to plan actions for embodied tasks.

However, open-ended environments are characterized by diverse and evolving task-relevant objects/relations/states that cannot be exhaustively pre-specified. In such embodied environments that involve intricate object properties and long sequences of interdependent actions, VLMs without task-specific training often generate ungrounded or hallucinated actions that violate causal logic (Wu et al., 2024; Huang et al., 2022b; Ahn et al., 2022; Liang et al., 2022; Wong et al., 2023; Huang et al., 2022a). These missteps tend to accumulate over time, ultimately undermining the reliability of planning.

To address this, recent approaches incorporate classical symbolic planners alongside large models by generating symbolic representations of the planning problem (Fung et al., 2025; Xiong et al., 2025; Dwivedi et al., 2023; Zhang et al., 2023; Silver et al., 2022), in order to mitigate hallucinated actions and causal violations. Symbolic representations could be expressed using the Planning Domain Definition Language (PDDL) (Aeronautiques et al., 1998; Zhi-Xuan, 2022), which separates a planning problem into a domain file that defines predicates, action schemas and transition rules, and a problem file that specifies objects, initial states, and goal conditions.

Nonetheless, as illustrated in Figure 1, these attempts to integrate large models with classical planners have revealed two challenges in modeling of symbolic representations and long-horizon planning: **(1) Incomplete and Static Symbolic World**: At the perception stage, existing methods (Liu et al., 2023; Migimatsu & Bohg, 2022) construct the symbolic world by using a VLM to ground observations and instructions into symbolic representations (objects and their symbolic states/relations in the PDDL problem). Yet in open-ended environments, many critical states and affordances are action-dependent and only revealed through interaction, making one-shot symbolic grounding inevitably incomplete. **(2) Non-adaptive Symbolic Knowledge**: While prior methods may incorporate feedback or iterative refinement, the resulting experience is often recorded

---

[*]Equal contribution  [1]Zhejiang University, Hangzhou, China.  [2]Lanzhou University, Lanzhou, China.  Correspondence to: Juncheng Li <junchengli@zju.edu.cn>.

*Proceedings of the 43rd International Conference on Machine Learning*, Seoul, South Korea. PMLR 306, 2026. Copyright 2026 by the author(s).

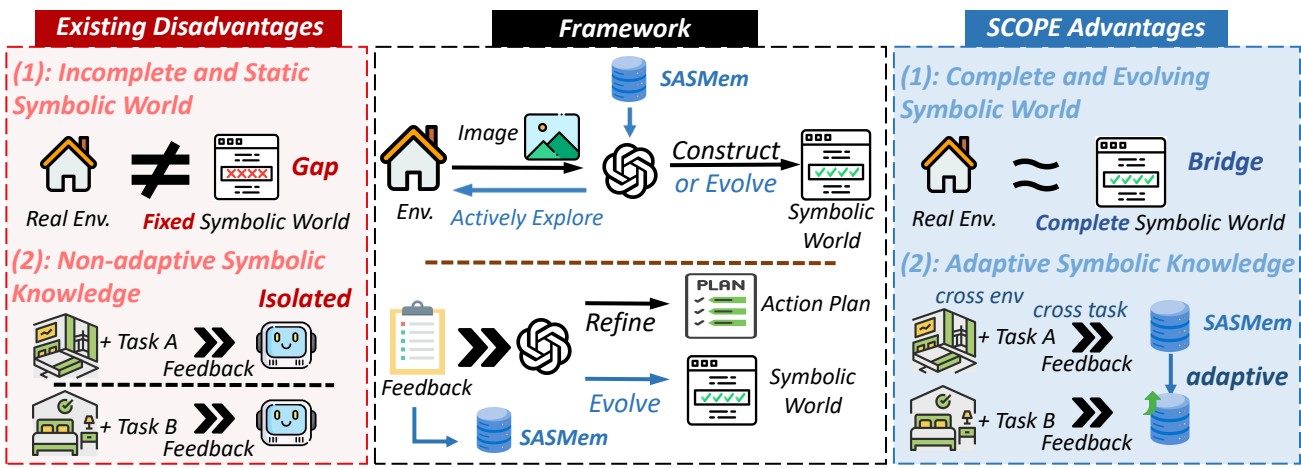

*Figure 1.* Comparison of existing method and SCOPE. In the middle illustration, **black arrows** denote the pipeline used by existing methods; **blue arrows** highlight **SCOPE** extensions.

as task-specific traces and is not often distilled into symbolic-friendly knowledge that is aligned with symbolic planning. As a consequence, the stored information has limited density and transferability, providing weaker cross-task and cross-scene support for symbolic planning and symbolic world refinement (Sarch et al., 2024; Zhou et al., 2024).

Inspired by the paradigm of reasoning with iterative symbolic correction (Zhou et al., 2024; Mao et al., 2019; Zhu et al., 2021), we propose **SCOPE**, a self-adaptive symbolic planning framework that constructs the symbolic world from perception and evolves the symbolic world and refines action plans through feedback from symbolic validation and real execution. Specifically, as shown in Figure 2, **SCOPE** evolves the symbolic world as the agent actively explores the open-ended environment. Through the Symbolic Execution Simulator (**SESim**), it could validate the logical consistency between the plan and the symbolic world, and generate the feedback from both symbolic validation and real execution to achieve grounded reasoning and drive exploration to construct a **complete and evolving symbolic world**. In addition, feedback is distilled into a Self-Adaptive Symbolic Memory (**SASMem**), enabling **adaptive symbolic knowledge** that enhances grounding across tasks and adaptability across diverse embodied environments.

Importantly, the two modules in **SCOPE** form a tightly coupled closed-loop framework that continually improves over time. The SASMem distills feedback into symbolic knowledge that enhances VLM's capability to generate more accurate symbolic worlds and action plans in future tasks. In turn, the SESim performs symbolic validation and real execution. Then, it provides feedback not only for plan refining but also for updating SASMem.

Extensive experiments validate that evolving symbolic world representations and abstracting reusable symbolic knowledge across tasks and across diverse embodied environments, enable strong continual adaptation, its step success rate improves by 14.4% throughout a continual run. Furthermore, in open-ended environments, this self-adaptive behavior requires fewer symbolic refinements and exhibits improved robustness compared to the NESYC baseline (Choi et al., 2025), achieving an average increase of 11.3% in task success rate across all evaluated open-ended tasks. In general, our contributions are as follows:

- **SCOPE** establishes a self-adaptive symbolic planning framework that evolves the symbolic world over time, providing a foundation for grounded planning within symbolic representations.

- Within **SCOPE**, the SESim integrates symbolic validation and real execution to narrow the representation gap between the symbolic world and the open-ended environment, and support more grounded action plans.

- **SCOPE** further incorporates the SASMem, which distills reusable feedback into action reasoning and world modeling knowledge aligned with symbolic planning, enabling cross-task grounding and adaptability across diverse embodied scenarios.

## 2. Related Works

### 2.1. Planning with Large Pretrained Models and Symbolic Feedback

Recent efforts have investigated enhancing the planning capabilities of Large Pretrained Models through prompts and symbolic feedback (Akakzia et al., 2020; Jiang et al., 2019a; Mirchandani et al., 2021; Silver et al., 2023; Srinivas et al., 2018; Nair & Finn, 2019; Choi et al., 2025). For example, ProgPrompt (Singh et al., 2023) proposes a programmatic

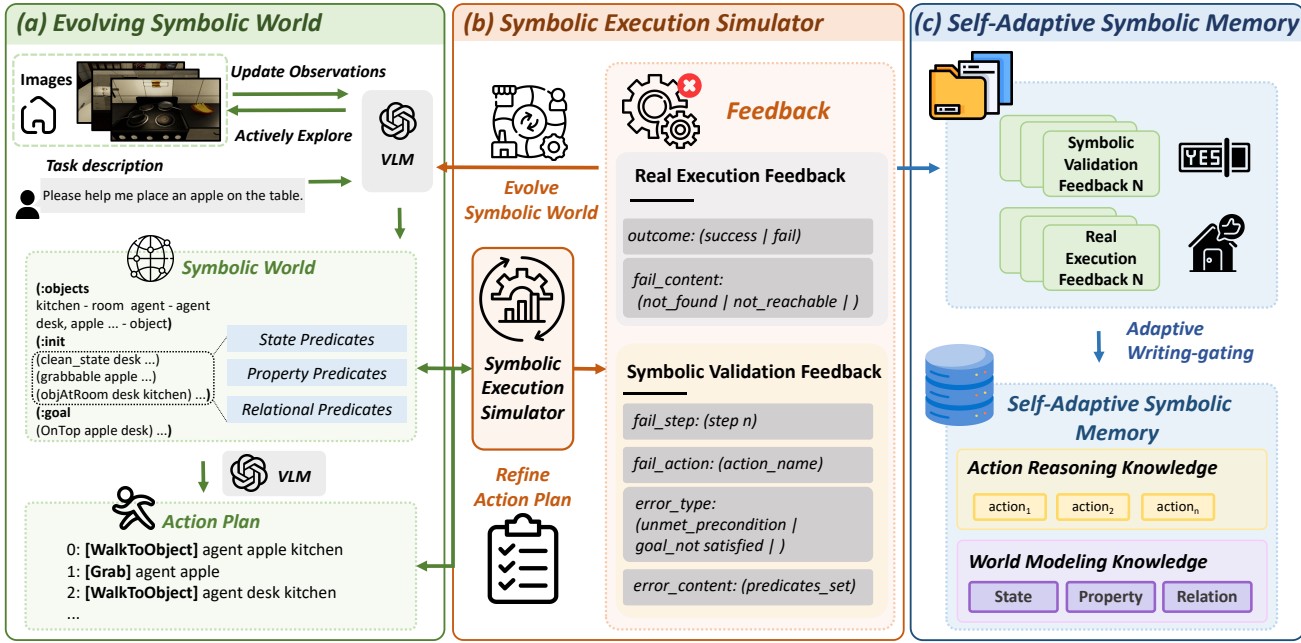

*Figure 2.* The overview of our framework. (a) Evolving Symbolic World: The agent actively explores the environment to refine the symbolic world, represented in PDDL problem file. Based on this symbolic world, the VLM generates a symbolic action plan for the embodied task. (b) Symbolic Execution Simulator: The generated plan is validated within the symbolic world using the PDDL validator and executed in the environment. SESim integrates feedback from symbolic validation and real execution to evolve the symbolic world and refine the action plan, narrowing the gap between symbolic and real environments. (c) Self-Adaptive Symbolic Memory: Feedback is distilled into abstract knowledge of action planning and world modeling. This knowledge is retrieved on demand to provide targeted hints for plan/world refinement.

prompt format that embeds executable examples and action templates in the LLM input. ISR-LLM (Zhou et al., 2024) further improves symbolic consistency through an iterative self-refinement process: Employ a classical validator to refine the LLM-generated plan over multiple rounds. This symbolic feedback loop increases plan feasibility while maintaining generalization from language.

However, these methods mainly focus on refining action plans for a single task and do not accumulate symbolic feedback over time. In contrast, **SCOPE** distills feedback into reusable symbolic knowledge that improves both symbolic world modeling and action reasoning across tasks and environments, enabling more robust and adaptive planning.

**2.2. Symbolic Modeling for Embodied Planning**

PDDL-based symbolic abstraction employs LLMs to translate natural language into symbolic domains and problem definitions, which are then solved using classical planners (Liu et al., 2023; Hu et al., 2025; Zhang et al., 2025; Jiang et al., 2019b). However, symbolic representations in prior works are typically static and incomplete. They are often handcrafted by experts, extracted from simulators using rule-based templates, or generated once from VLM perception at task initialization. Such fixed abstractions are insufficient for open-ended and dynamic environments that de-

mand continual adaptation. In contrast, our method evolves the symbolic world over time using feedback from symbolic validation and real execution. This enables more grounded and adaptive planning in complex embodied tasks.

# 3. Method

We propose **SCOPE**, a self-adaptive symbolic planning framework that enhances symbolic world modeling and long-horizon planning in embodied tasks. As shown in Figure 2, **SCOPE** actively explores the open-ended environment, constructs the symbolic world and generates an action plan (Section 3.1). The Symbolic Execution Simulator then provides feedback to evolve the symbolic world and refine the action plan (Section 3.2). **SCOPE** further distills feedback into symbolic knowledge, to enhance long-horizon planning and symbolic world modeling across tasks and scenarios (Section 3.3).

## 3.1. Evolving Symbolic World

In the open-ended environment, a central challenge in integrating large models with classical planners is how the symbolic world, the symbolic representations of planning problem, can accurately describe the real environment without gaps, thereby supporting classical planner verification

or planning. This requires symbolic world to expand along with the environment for its open-ended nature. We address this challenge with the evolving symbolic world.

As illustrated in Figure 2, the first stage of our framework constructs an initial symbolic world $\mathcal{W}_{symbol}$ from raw visual observations and task description. Within $\mathcal{W}_{symbol}$, objects are described by several types of predicates: (1) State Predicates, which capture dynamic properties of objects; (2) Property Predicates, which describe inherent object attributes; (3) Relational predicates, which express spatial and logical relationships between objects.

Based on the current symbolic world, the agent decides to either (1) explore the environment to identify additional object representations or interact with objects to enhance the understanding of object attributes. Both contribute to evolving symbolic world; or (2) generate a symbolic action plan $\mathcal{P}_A = (a_1, a_2, \ldots, a_n)$, follows a PDDL format, conditioned on $\mathcal{W}_{symbol}$.

### 3.2. Symbolic Execution Simulator

A complete symbolic world is not derived solely from visual perception, but also from active interaction with the environment. For example, discovering that an object is not graspable only after a failed attempt to grab it. This embodied feedback reveals latent properties that are often inaccessible to vision alone. To address this, Symbolic Execution Simulator(SESim) integrates feedback from both symbolic validation and real execution to reduce the gap between the symbolic world and the open-ended environment.

As shown in Figure 2, following the construction of the symbolic world $\mathcal{W}_{symbol}$ and the generation of the initial action plan $\mathcal{P}_A$ in Section 3.1, **SCOPE** employs the Symbolic Execution Simulator to perform symbolic validation within symbolic world.

The SESim operates in two stages: (1) Symbolic Validation. The SESim utilizes the classical PDDL validator to verify the logical consistency of $\mathcal{P}_A$ within the current symbolic world $\mathcal{W}_{symbol}$, generating symbolic validation feedback $\mathcal{F}_{symbol}$. Feedback indicates logical issues such as undefined objects, unmet preconditions, or incomplete goal satisfaction. $\mathcal{F}_{symbol}$ is interpretable, and supports to refine action plan. (2) Real Execution. The SESim executes the action plan in the environment, generating real execution feedback $\mathcal{F}_{real}$. Even when the action plan passes symbolic validation, real execution failures can expose gaps between the symbolic world and open-ended environment, such as missing property predicates or incorrect relationships.

These structured feedback fields also trigger *symbolic retrieval* from SASMem. SESim forms a symbol-aligned query from $\mathcal{F}_{symbol}$ and $\mathcal{F}_{real}$ (failure signature and predicate-level mismatch evidence), retrieves the most relevant ac-

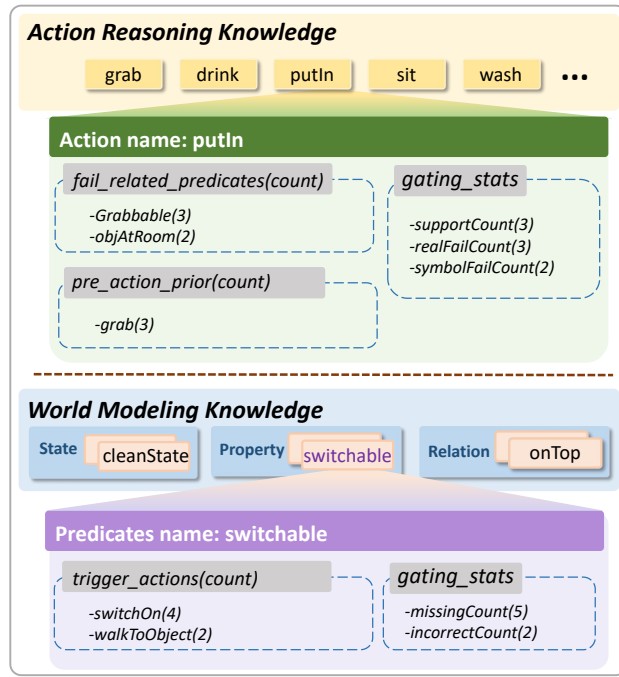

*Figure 3.* Overview of SASMem, illustrating its structure and symbolic knowledge components.

tion/world entries, and aggregates them as $\mathcal{F}_{SASMem}$ to guide localized plan/world updates:

$$\mathcal{P}'_A, \mathcal{W}'_{symbol} = \text{VLM}(\mathcal{P}_A, \mathcal{F}_{symbol}, \mathcal{F}_{real}, \mathcal{F}_{SASMem})$$

SESim iterates this process, until validation succeeds or a budget is reached, thereby increasing robustness in the open-ended environment. By leveraging complementary symbolic validation and grounded feedback, **SCOPE** reduces representational gaps and inconsistencies, enabling more accurate symbolic world modeling and more reliable long-horizon planning.

### 3.3. Self-Adaptive Symbolic Memory

Successfully completing an embodied task in a specific environment, the agent can acquire valuable experience, such as understanding the logical structure of actions, reasoning over plans, and modeling the environment. A natural question arises: can such experience generalize to more complex tasks and environments? To support generalizable symbolic planning across diverse tasks and environments, **SCOPE** incorporates the Self-Adaptive Symbolic Memory (SASMem) , a structured memory that distills predicate-level knowledge about *action reasoning* and *world modeling*.

**Action reasoning knowledge** captures reusable regularities of action failures and successes: for each action, it summarizes the most frequently implicated symbolic conditions (e.g., missing preconditions revealed by symbolic

validation) and the common successful local action patterns (i.e., which prerequisite steps tend to precede the action in successful plans).

**World modeling knowledge** captures persistent gaps between the symbolic world and the real environment at the predicate level. It records which predicates are commonly missing or mis-specified under grounded interaction evidence, and which interaction actions most often surface such evidence, thereby providing targeted guidance for evolving the symbolic world.

SASMem does not store raw experience directly, nor does it rely on the VLM to produce a summary; instead, it applies rule-based distillation over structured $\mathcal{F}_{\text{symbol}}$, $\mathcal{F}_{\text{real}}$ and experience to generate predicate-level memory entries. Memory updates follow a write-gating rule: an injected entry is reinforced only when it resolves the same failure pattern or pushes the first failure to a later step, and is otherwise down-weighted, with symbolic infeasibility and real-execution mismatches treated distinctly. This mechanism prevents unbounded accumulation, automatically suppresses noisy memories, and yields a continually improving memory that enhances long-horizon grounding and adaptation across tasks and environments.

## 4. Experiments

Our goal is to evaluate whether **SCOPE** improves symbolic world modeling and long-horizon planning in embodied tasks under dynamic and open-ended perception. In this section, we first present the experimental setup (Section 4.1), followed by comprehensive evaluations on two key fronts: (1) grounded planning under environment perturbations (Section 4.2), (2) robust symbolic world modeling (Section 4.3). In addition, we conduct an in-depth analysis (Section 4.4), which includes: qualitative examples, ablations on the Self-Adaptive Symbolic Memory (SASMem) and Symbolic Execution Simulator (SESim), and extensive evaluations across environments and tasks to assess the generalization and robustness of the method.

### 4.1. Experiment Setting

**Environments.** We evaluate on two embodied benchmarks, *VirtualHome* (Puig et al., 2018) and *ALFRED* (Shridhar et al., 2020). For each benchmark, we construct evaluation settings along two axes: **scene complexity** (*basic* vs. *complex*) and **environment settings** (*static*, *dynamic*, *open-ended*).

**Scene complexity.** *Basic* scenes feature simpler layouts with fewer interactive objects and shorter dependency chains, whereas *complex* scenes contain larger layouts with richer interactions and longer-horizon plans.

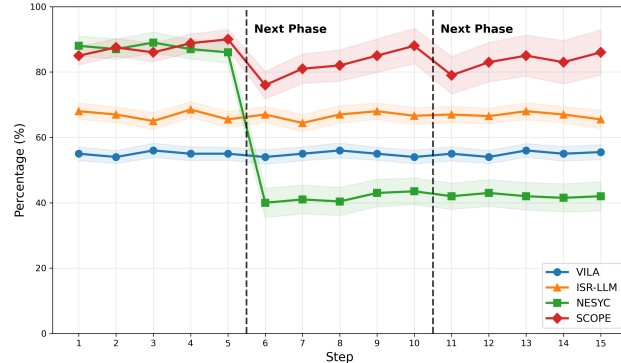

*(a)* VirtualHome: Per-steps StepSR on Open-ended step stream symbolic world

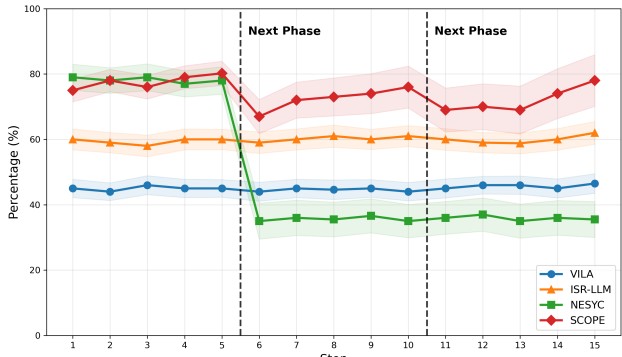

*(b)* ALFRED: Per-steps StepSR on Open-ended step stream

*Figure 4.* Symbolic world evolution in open-ended settings.

**Environment settings.** *Static* settings keep the environment state unchanged unless affected by the agent's actions. *Dynamic* settings introduce state changes during an episode, requiring the agent to re-ground and re-plan under non-stationary conditions. *Open-ended* settings stream a sequence of steps in which new task-required affordances or state dependencies are introduced in phases over time, requiring continual online adaptation and knowledge carry-over across tasks. The detailed construction protocol for these splits and changes is provided in the Appendix.

**Baselines.** We implement three baselines, categorized into two groups: (i) VLM-based planners: *VILA* (Hu et al., 2023); (ii) neuro-symbolic methods with symbolic feedback/tools: *ISR-LLM* (Zhou et al., 2024)(we use a faithful VLM-adapted re-implementation of the original LLM-based method for embodied inputs) and *NESYC* (Choi et al., 2025). For fair comparison, we use the same backbone model GPT-4o, for all VLM components in the methods.

**Evaluation Metrics.** We report metrics for both symbolic world quality and planning performance. For planning, we follow common practice and report *SR* (%), the percentage of tasks successfully completed, and *GC* (%), the suc-

*Table 1.* Planning performance (SR/GC, %) on VirtualHome and ALFRED under basic/complex scenes and static/dynamic/open-ended settings. For *open-ended* evaluation, SR/GC are averaged over the entire step stream

| VirtualHome | | Static | | Dynamic | | Open-ended | |
|---|---|---|---|---|---|---|---|
| Methods | Planner | SR | GC | SR | GC | SR | GC |
| Basic scenes | | | | | | | |
| VILA | VLM | $55.4 \pm 1.2$ | $69.2 \pm 1.9$ | $52.3 \pm 1.3$ | $55.1 \pm 1.8$ | $47.8 \pm 1.4$ | $52.4 \pm 1.9$ |
| ISR-LLM | VLM | $65.8 \pm 1.1$ | $70.5 \pm 1.6$ | $61.7 \pm 1.2$ | $67.2 \pm 1.5$ | $58.1 \pm 1.3$ | $62.6 \pm 1.7$ |
| NESYC | Classical planner | $\mathbf{90.1 \pm 0.7}$ | $\mathbf{92.7 \pm 0.6}$ | $77.8 \pm 1.0$ | $80.1 \pm 1.2$ | $56.3 \pm 1.0$ | $58.9 \pm 1.1$ |
| **SCOPE** | VLM | $88.2 \pm 0.8$ | $89.4 \pm 0.9$ | $\mathbf{85.4 \pm 0.9}$ | $\mathbf{86.9 \pm 0.8}$ | $\mathbf{87.6 \pm 0.8}$ | $\mathbf{89.2 \pm 0.9}$ |
| Complex scenes | | | | | | | |
| VILA | VLM | $45.2 \pm 1.4$ | $46.8 \pm 2.1$ | $40.6 \pm 1.5$ | $41.4 \pm 2.0$ | $38.1 \pm 1.6$ | $40.7 \pm 2.2$ |
| ISR-LLM | VLM | $62.3 \pm 1.3$ | $67.1 \pm 1.8$ | $58.2 \pm 1.4$ | $62.8 \pm 1.7$ | $54.5 \pm 1.5$ | $56.0 \pm 1.9$ |
| NESYC | Classical planner | $84.3 \pm 0.8$ | $85.5 \pm 0.7$ | $72.4 \pm 1.2$ | $77.7 \pm 1.4$ | $50.9 \pm 1.2$ | $53.5 \pm 1.3$ |
| **SCOPE** | VLM | $\mathbf{86.5 \pm 1.0}$ | $\mathbf{88.8 \pm 1.1}$ | $\mathbf{82.1 \pm 1.0}$ | $\mathbf{85.6 \pm 0.9}$ | $\mathbf{79.4 \pm 0.9}$ | $\mathbf{82.0 \pm 1.0}$ |

| ALFRED | | Static | | Dynamic | | Open-ended | |
|---|---|---|---|---|---|---|---|
| Methods | Planner | SR | GC | SR | GC | SR | GC |
| Basic scenes | | | | | | | |
| VILA | VLM | $50.7 \pm 1.3$ | $57.4 \pm 2.0$ | $47.2 \pm 1.4$ | $50.8 \pm 1.9$ | $44.3 \pm 1.5$ | $45.9 \pm 2.0$ |
| ISR-LLM | VLM | $63.2 \pm 1.2$ | $79.1 \pm 1.7$ | $58.6 \pm 1.3$ | $61.5 \pm 1.6$ | $61.8 \pm 1.4$ | $64.3 \pm 1.8$ |
| NESYC | Classical planner | $\mathbf{88.4 \pm 0.7}$ | $\mathbf{90.1 \pm 0.6}$ | $71.7 \pm 1.1$ | $75.2 \pm 1.3$ | $50.1 \pm 1.1$ | $53.5 \pm 1.2$ |
| **SCOPE** | VLM | $85.8 \pm 0.9$ | $89.1 \pm 1.0$ | $\mathbf{83.1 \pm 0.9}$ | $\mathbf{85.7 \pm 0.8}$ | $\mathbf{80.8 \pm 0.8}$ | $\mathbf{84.4 \pm 0.9}$ |
| Complex scenes | | | | | | | |
| VILA | VLM | $36.5 \pm 1.5$ | $43.2 \pm 2.2$ | $33.8 \pm 1.6$ | $37.1 \pm 2.1$ | $30.9 \pm 1.7$ | $32.2 \pm 2.3$ |
| ISR-LLM | VLM | $59.8 \pm 1.4$ | $63.6 \pm 1.9$ | $55.3 \pm 1.5$ | $58.0 \pm 1.8$ | $54.4 \pm 1.6$ | $56.8 \pm 2.0$ |
| NESYC | Classical planner | $\mathbf{82.8 \pm 0.8}$ | $\mathbf{83.6 \pm 0.7}$ | $64.3 \pm 1.3$ | $67.8 \pm 1.5$ | $51.7 \pm 1.3$ | $52.1 \pm 1.4$ |
| **SCOPE** | VLM | $78.4 \pm 1.1$ | $80.5 \pm 1.2$ | $\mathbf{76.9 \pm 1.0}$ | $\mathbf{78.4 \pm 0.9}$ | $\mathbf{75.6 \pm 1.0}$ | $\mathbf{77.2 \pm 1.1}$ |

cess rate of individual goal conditions. For open-ended step-stream evaluation, we additionally report *StepSR* (%), Step Success Rate, defined as the step-level success rate of executed actions in the environment: it measures the percentage of action steps that are successfully carried out, averaged over the entire step stream. For symbolic world modeling, we measure *ClassicalSR* (%), the fraction of tasks whose constructed symbolic world is sufficiently complete/consistent for a classical planner to produce a valid plan; *SymRecall* (%), the recall of benchmark-/expert-provided ground-truth task-relevant predicates in the symbolic world.

### 4.2. Planning under Environment Settings

**Robust planning across settings.** Table 1 compares long-horizon planning under *static*, *dynamic*, and *open-ended* settings. In *static* environments, the symbolic world is relatively complete and stable, so methods that rely on a classical planner as the planner can leverage exact symbolic search to produce causally grounded action plans. However, this advantage diminishes in *dynamic* settings, where non-stationary state changes quickly make the initial sym-

bolic snapshot stale. Once key predicates become missing or outdated, classical planners tend to fail and crucially cannot effectively inform the VLM which specific predicates or state facts are absent. By contrast, the VLM can still propose actions, while symbolic validation and execution feedback can be used to pinpoint violated preconditions and mismatches, enabling targeted correction. This brittleness is further amplified in *open-ended* episode streams, where newly emerging affordances and state dependencies systematically exceed the initial symbolic coverage.

**Open-ended Step-Stream Results.** We report *StepSR* (%), *Step Success Rate*, which measures whether each executed action step succeeds in the environment over the open-ended stream. *VILA* (VLM-only planner) remains largely flat across the stream, with only minor fluctuations, yet consistently exhibits weak step executability. *ISR-LLM* shows similarly limited variation across phases; while it benefits from validator-provided symbolic error signals, symbolic feedback alone is often insufficient to bridge the symbolic–real gap under continual novelty. In contrast, *NESYC* (classical planner) performs strongly early on but degrades sharply after phase transitions, indicating brittleness when newly in-

*Table 2.* Symbolic world quality measured by *ClassicalSR* and *SymRecall* on both benchmarks, averaged over *open-ended* settings.

| Method | Evolve World? | ClassicalSR | | SymRecall | |
|---|---|---|---|---|---|
| | | ALFRED | VirtualHome | ALFRED | VirtualHome |
| Basic scenes | | | | | |
| **SCOPE** | ✓ | **94.8 ± 0.8** | **95.8 ± 0.7** | **69.2 ± 2.8** | **65.5 ± 3.1** |
| w/o SESim | ✗ | 72.3 ± 3.9 | 74.2 ± 1.2 | 52.1 ± 3.5 | 53.3 ± 3.7 |
| w/o SASMem | ✓ | 80.2 ± 1.8 | 85.3 ± 0.9 | 54.1 ± 3.2 | 50.2 ± 3.4 |
| w/o SESim & SASMem | ✗ | 70.0 ± 5.4 | 72.8 ± 1.6 | 50.1 ± 4.8 | 48.1 ± 4.9 |
| Complex scenes | | | | | |
| **SCOPE** | ✓ | **89.0 ± 1.0** | **90.8 ± 0.8** | **52.3 ± 4.2** | **64.1 ± 3.5** |
| w/o SESim | ✗ | 64.2 ± 5.1 | 74.3 ± 3.3 | 45.4 ± 4.8 | 51.2 ± 4.6 |
| w/o SASMem | ✓ | 70.2 ± 1.4 | 75.6 ± 1.5 | 44.9 ± 4.7 | 48.9 ± 4.5 |
| w/o SESim & SASMem | ✗ | 45.2 ± 5.8 | 61.6 ± 5.9 | 37.6 ± 5.5 | 40.7 ± 5.0 |

troduced affordances or dependencies exceed the initial symbolic coverage. **SCOPE** achieves the most robust behavior: although it may exhibit transient drops near phase boundaries, it rapidly recovers in subsequent steps, demonstrating effective online adaptation under open-ended changes.

**Planning over long horizons and increasing scene complexity.** We further compare *basic* versus *complex* scenes to stress-test scalability under longer action horizons and richer object interactions. While baselines tend to degrade as complexity increases due to compounding grounding errors and limited symbolic coverage, **SCOPE** remains more robust by continuously updating symbolic predicates and revising plan steps based on newly observed evidence during execution. Overall, these results suggest that **SCOPE** not only adapts to changing and open-ended environments, but also better sustains coherent and grounded long-horizon planning as scene complexity grows.

### 4.3. Symbolic World Evaluation

Symbolic worlds constructed directly by a VLM in *open-ended* settings are often incomplete or inconsistent, resulting in low *ClassicalSR* and fragile long-horizon planning (Table 2). Methods without symbolic world evolution remain constrained by a static world model and struggle to keep pace with newly introduced affordances and state dependencies over the stream. In contrast, *SESim* leverages feedback from symbolic validation and real execution to iteratively evolve the symbolic world, correcting missing predicates and faulty preconditions, which substantially improves *ClassicalSR*, especially in complex scenes. We further assess semantic alignment with *SymRecall*: one-shot perception commonly misses object attributes and relations, whereas feedback-driven refinement uncovers latent properties revealed through interaction (e.g., non-graspable objects), underscoring the importance of interaction-informed world evolution under continual novelty.

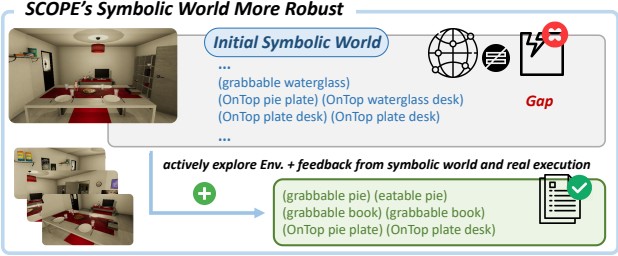

*Figure 5.* Symbolic world evolution example.

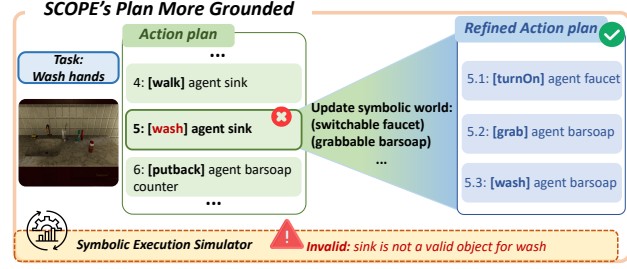

*Figure 6.* Action plan refinement example.

### 4.4. In-Depth Analysis and Generalization

**Qualitative Analysis.** We qualitatively examine how **SCOPE** enhances symbolic modeling and planning under environment perturbations, shown in Figure 5 and Figure 6. Compared to direct VLM outputs, **SCOPE** produces more complete symbolic world by feedback. This leads to better recovery of missing object attributes and relational representations. In planning, symbolic validation allows **SCOPE** to detect and resolve causal gaps, resulting in more grounded and long-horizon action plan.

**Analysis of Symbolic Execution Simulator.** We assess the respective roles of symbolic validation and real execution feedback within SESim (Table 3). Using only real execution

*Table 3.* Ablation on SESim feedback sources on *VirtualHome*. Impact of using real execution feedback only, symbolic validation only, or both on *ClassicalSR*, *SymRecall*, and *GC*.

| Method | ClassicalSR | SymRecall | GC |
|---|---|---|---|
| SESim(Real exec. only) | $80.2 \pm 1.8$ | $55.6 \pm 2.4$ | $81.6 \pm 1.7$ |
| SESim(Validation only) | $86.5 \pm 1.5$ | $57.1 \pm 2.1$ | $80.3 \pm 1.6$ |
| SESim(Full) | $92.6 \pm 1.2$ | $62.6 \pm 1.8$ | $82.1 \pm 1.3$ |

*Table 4.* Average plan/world update counts on *VirtualHome* under different memory designs (*#PlanRefine*/*#EnvEvolve*).

| Method | #PlanRefine | #EnvEvolve |
|---|---|---|
| RawMem | $6.51 \pm 0.45$ | $3.91 \pm 0.33$ |
| SummMem | $4.21 \pm 0.35$ | $2.02 \pm 0.24$ |
| SASMem | $2.18 \pm 0.28$ | $0.86 \pm 0.18$ |

feedback allows the agent to detect that an action has failed and that a mismatch exists between the symbolic world and the real environment, but it cannot identify the underlying cause—whether the failure arises from a missing object or from incorrect object properties. Conversely, relying on symbolic validation exposes logical error in the plan, such as unmet preconditions or missing symbolic entities. However, evolving the symbolic world based only on symbolic validation can inadvertently enlarge the gap between symbolic representations and open-ended environment. Combined, they provide complementary feedback that jointly guide more accurate symbolic world evolution and more robust planning.

**Analysis of Self-Adaptive Symbolic Memory.** We analyze the impact of SASMem by comparing it with two generic memory-augmented baselines, *RawMem* and *SummMem*, which store and retrieve past experience to augment the VLM prompt without structured, predicate-level distillation (Table 4). Specifically, *RawMem* retrieves raw traces, while *SummMem* retrieves VLM-produced summaries. In contrast, SASMem organizes feedback from symbolic validation and real execution in a predicate-level, symbol-aligned form with higher information density, enabling more efficient and precise guidance—reflected by fewer plan refinements and symbolic world updates.

**Symbolic Hallucination Evaluation.** We measure *SymHalluc* (%), defined as the fraction of introduced/refined task-relevant predicates that are inconsistent with the ground truth, under basic and complex environments (Table 5). SASMem contributes to symbolic grounding through adaptive knowledge prompting, its ability to suppress hallucinated predicates remains limited, particularly when symbolic world diverge from the open-ended environments. In contrast, SESim plays a key role in mitigating symbolic hallucination by leveraging feedback from symbolic validation and real execution. This feedback helps identify and

*Table 5.* Impact of SESim and SASMem on *SymHalluc*(%).

| Method | SymHalluc (%) | |
|---|---|---|
| | VirtualHome | ALFRED |
| SCOPE | $\mathbf{3.68 \pm 0.6}$ | $\mathbf{5.50 \pm 0.8}$ |
| w/o SASMem | $5.72 \pm 1.3$ | $7.91 \pm 1.4$ |
| w/o SESim | $9.88 \pm 1.9$ | $10.43 \pm 2.0$ |
| w/o SASMem & SESim | $14.01 \pm 2.6$ | $16.28 \pm 3.1$ |

*Table 6.* Impact of different backbones on planning and symbolic world modeling.

| Method | SR | GC | ClassicalSR | SymRecall |
|---|---|---|---|---|
| **Qwen3-VL-8B** | | | | |
| ISR-LLM | $57.7 \pm 1.1$ | $63.2 \pm 1.4$ | $67.2 \pm 2.8$ | $53.1 \pm 2.7$ |
| SCOPE | $76.8 \pm 2.3$ | $78.7 \pm 1.7$ | $89.3 \pm 2.5$ | $59.6 \pm 3.2$ |
| **GPT-4o** | | | | |
| ISR-LLM | $65.2 \pm 2.2$ | $70.5 \pm 1.8$ | $70.4 \pm 2.7$ | $52.5 \pm 2.2$ |
| SCOPE | $86.5 \pm 1.3$ | $88.5 \pm 2.1$ | $96.5 \pm 1.5$ | $63.5 \pm 1.3$ |

revise symbolic hallucination, progressively narrowing the gap between the symbolic world and the real environment.

**Impact of Different VLMs.** We further examine backbone sensitivity on *VirtualHome*, and observe consistent trends across benchmarks. Table 6 shows that **SCOPE** consistently outperforms ISR-LLM across both backbones on planning performance (SR/GC) as well as symbolic world quality (ClassicalSR/SymRecall). Notably, the advantage persists with the smaller backbone (Qwen3-VL-8B), suggesting that the improvements are driven by our feedback-grounded closed-loop—i.e., iteratively evolving the symbolic world and refining plans using symbolic validation and real execution—rather than relying solely on model scale. When switching to a stronger backbone (GPT-4o), both methods improve in absolute terms, yet **SCOPE** retains a clear margin, indicating that the proposed refinement and memory distillation mechanisms generalize across backbone choices and further benefit from increased model capacity.

## 5. Conclusion

We present **SCOPE**, a self-adaptive symbolic planning framework that evolves symbolic world representations through feedback from both symbolic validation and real execution. By closing the loop between perception, reasoning, and interaction, **SCOPE** addresses the limitations of static symbolic modeling and non-adaptive symbolic knowledge in open-ended environments, providing a general way to integrate large models with classical planners. Experiments demonstrate that evolving symbolic representations leads to more grounded long-horizon planning, and better generalization across tasks and environments.

## Acknowledgements

This work was supported by the National Key Research and Development Program of China (2025ZD0123100), National Natural Science Foundation of China (62436007), Zhejiang NSF (LQK26F020001), Key R&D Program of Zhejiang (2026SDXT005), Ningbo Yongjiang Talent Introduction Programme (2024A-401-G), Zhejiang University Education Foundation Qizhen Scholar Foundation.

## Impact Statement

This paper presents work whose goal is to advance the field of Machine Learning. There are many potential societal consequences of our work, none which we feel must be specifically highlighted here.

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

# Appendix

# A. Limitations and Future Discussions

While **SCOPE** demonstrates improvements in symbolic planning and long-horizon planning, there are several limitations that warrant further exploration. In this section, we discuss the current limitations of our framework and outline potential directions for future work.

## A.1. Limitations

**Real-World Execution Feedback.** In a simulator, real execution feedback is directly provided by the program, such as when an action cannot be executed or when an object is not found. However, in the real world, feedback from real execution is more difficult to describe. Unlike in a simulated environment where actions and object states are explicitly programmed, real-world feedback involves sensory inputs, environmental interactions, and physical constraints that are not always easily translated into direct feedback or error messages. This complexity requires sophisticated sensor integration and real-time processing to accurately reflect the outcomes of actions.

## A.2. Future Directions

**Integration with Reinforcement Learning (RL).** Future work could explore integrating **SCOPE** with RL to enhance the agent's ability to learn policies for symbolic task planning. By combining symbolic planning with RL, the agent may not only refine its symbolic world representations but also optimize long-horizon planning by exploring more complex policies and rewards.

**Expanding to Multi-Agent Environments. SCOPE** could be extended to multi-agent environments, where the complexity of planning increases significantly. PDDL has the ability to model complex multi-agent interactions, and integrating it into **SCOPE** could enable the system to handle planning scenarios that involve multiple agents with distinct goals and constraints. This would make **SCOPE** more applicable to collaborative robotics and other multi-agent systems.

# B. Evaluation Setup

## B.1. Task Category Specification

We summarize the average symbolic action length and the number of involved environment objects for each task category in Table 7.

**VirtualHome.** We use a subset of $n = 186$ tasks randomly sampled from the *VirtualHome* dataset. To assess the generalization of symbolic planning across diverse tasks and semantic domains, we further organize these tasks into four

*Table 7.* Average number of symbolic actions and objects in the environment involved in the scenes in *VirtualHome* and *ALFRED*.

| TaskName | #Actions (avg.) | #Objects (avg.) |
|---|---|---|
| VirtualHome | | |
| Personal Hygiene | 4.3 | 51.6 |
| Food and Dining | 7.2 | 66.8 |
| Housekeeping | 6.3 | 54.5 |
| Leisure | 5.3 | 33.7 |
| ALFRED | | |
| Pick & Place | 16.2 | 42.2 |
| Examine in Light | 8.7 | 37.7 |
| Clean & Place | 18.1 | 57.1 |
| Heat & Place | 11.2 | 69.6 |
| Cool & Place | 20.4 | 80.6 |
| Pick Two & Place | 14.8 | 56.8 |

manually defined categories based on their object dependencies. (1) *Personal Hygiene*: Brush teeth, Wash hands, Go to toilet (2) *Food and Dining*: Cook some food, Drink, Set up table, Put groceries in fridge (3) *Housekeeping*: Wash clothes, Wash dishes by hand, Wash dishes with dishwasher, Turn on lightswitch (4) *Leisure*: Watch TV, Change TV channel, Listen to music, Read book, Relax on sofa, Pick up phone, Browse internet

**ALFRED.** We use a subset of $n = 600$ tasks randomly sampled from the *ALFRED* dataset. These tasks are selected from the following six categories: (1) *Pick & Place*, (2) *Pick Two & Place*, (3) *Examine in Light*, (4) *Clean & Place*, (5) *Heat & Place*, and (6) *Cool & Place*. From each of these categories, we randomly select 100 tasks, ensuring a diverse set of scenarios to evaluate **SCOPE**. These tasks test the method's ability to handle complex action sequences and to manage multiple objects in open-ended environments.

## B.2. Open-ended Setting Construction

The open-ended setting is designed to evaluate adaptation when (i) the task distribution shifts over time and (ii) task-required affordances/state dependencies emerge beyond the agent's initial symbolic coverage. Unlike the static/dynamic settings that evaluate each task independently, the open-ended setting evaluates a stream of tasks with phased novelty, where knowledge can be carried over across tasks.

**Phase-based task stream.** We construct an ordered task stream $\{\tau_1, \tau_2, \ldots, \tau_K\}$ and partition it into $K = 3$ consecutive phases. Each phase corresponds to a task category (Appendix B.1) and is intended to introduce a new category-specific novelty. Within a phase, tasks are randomly permuted; across phases, the phase order is fixed for a given stream seed. For open-ended evaluation, we execute tasks sequentially and *do not reset* the agent's symbolic knowl-

*Table 8.* Schema of SASMem.

| Field | Description |
|---|---|
| **Action reasoning knowledge** | |
| action name | The target action that this entry describes. |
| fail related predicates | A multiset predicate count of predicates frequently implicated when *action_name* fails. |
| support count | Number of times retrieving this entry leads to the same failure pattern being resolved (or the first failure pushed to a later step). |
| real fail count | Number of times the plan still fails at real execution after using this entry. |
| symbol fail count | Number of times the plan still fails in symbolic validation after using this entry. |
| pre action prior | A sequence prior extracted from successful plans: we count which action most frequently occurs immediately before *action_name*. |
| **World modeling knowledge** | |
| predicate name | The predicate that this entry models. |
| trigger action | Action count indicating which interaction actions most often trigger grounded evidence or real failures related to *predicate_name* (typically derived from *fail_action*). |
| missing count | Number of times grounded execution evidence indicates *predicate_name* is required but missing from symbolic world. |
| incorrect count | Number of times grounded execution evidence indicates symbolic world contains an incorrect instance/value of *predicate_name*. |

edge/memory between tasks (the environment is reset per task as in standard evaluation, but the agent's internal state persists), enabling knowledge carry-over across the stream.

**Category-specific new action/affordance injection.** To simulate the introduction of novel actions in an open-ended manner, we associate each task category $c_k$ (the $k$-th phase) with a *category-specific* symbolic action template $a_{\text{new}}^{(k)}$ that represents a new affordance or a new state dependency required by tasks in that category. We implement this novelty by *gating* the availability of action schemas (and their predicate interfaces) across phases: at initialization (phase 1), the agent is only provided with a core set of generic actions (e.g., navigation/manipulation primitives and common interaction actions). At the beginning of phase $k$, the environment contains tasks whose successful completion *requires* at least one instance of $a_{\text{new}}^{(k)}$ (or its associated predicate-level dependency), while this action template is absent from the agent's initial symbolic coverage.

### B.3. Basic vs. Complex Environments.

For both *VirtualHome* and *ALFRED*, we further partition tasks into *basic* and *complex* environments in order to analyze how **SCOPE** behaves under different levels of environment complexity, as summarized in Table 10.

For *VirtualHome*, we define complexity based on the spatial extent of the scene explored by the agent. During execution,

we record the set of distinct rooms that the agent visits while completing the task. If the task can be accomplished within a single room, we label it as belonging to the *basic* environment. If the agent needs to visit multiple rooms to complete the task, we label it as belonging to the *complex* environment. The average symbolic action sequence length and the average number of environment objects involved per task for each category are summarized in Table 7.

For *ALFRED*, we define complexity based on the number of objects in the environment involved in the scenes. We label a task as *basic* if this number is at most 30, and as *complex* otherwise. The threshold of 30 is chosen such that both splits contain a sufficient number of tasks while the *complex* split exhibits significantly higher object cardinality.

### B.4. Evaluation Metrics

We provide precise definitions for all evaluation metrics used in our experiments:

- **Step Success Rate (StepSR)**: The step-level success rate of actions executed in the environment, averaged over the entire open-ended step stream:

$$\text{StepSR} = \frac{N_{\text{exec\_succ}}}{N_{\text{exec}}}$$

where $N_{\text{exec}}$ is the total number of action steps actually

*Table 9.* Structured feedback schema used by SESim.

| Field | Description |
|---|---|
| **Symbolic validation feedback** | |
| fail step | Index of the first failing step in the action sequence during symbolic validation. |
| fail action | Name of the failing action. |
| error type | Error category returned by the PDDL validator, e.g., *unmet_precondition*, *undefined_object*, *goal_not_satisfied*, ... |
| error content | Detailed error content. For *unmet_precondition*, it lists the missing predicate instances (e.g., *Graspable*). |
| **Real execution feedback** | |
| outcome | Execution outcome of the attempted plan in the environment: success or fail. |
| fail content | Failure reason reported by the environment e.g., *not_found*, *not_reachable*, ... |

*Table 10.* Average number of symbolic actions and distinct interactive objects per task, as well as the number of tasks in each group.

| task | #Actions (avg.) | #Objects (avg.) | #Tasks |
|---|---|---|---|
| **VirtualHome** | | | |
| basic | 5.3 | 34.7 | 57 |
| complex | 7.9 | 82.9 | 129 |
| **ALFRED** | | | |
| basic | 12.8 | 29.3 | 227 |
| complex | 15.9 | 68.9 | 273 |

executed in the environment over the step stream, and $N_{\text{exec\_succ}}$ is the number of those steps that succeed.

- **Classical Planner Support Rate (ClassicalSR)**: The percentage of tasks in which the symbolic world is sufficiently complete to allow a classical planner to generate a valid plan:

$$\text{ClassicalSR} = \frac{N_{\text{supported}}}{N_{\text{total}}}$$

where $N_{\text{supported}}$ is the number of tasks with planner-supported symbolic models, and $N_{\text{total}}$ is the total number of tasks.

- **Task-Relevant Predicate Recall (SymRecall)**: The percentage of ground-truth task-relevant predicates correctly recovered in the symbolic world:

$$\text{SymRecall} = \frac{|\mathcal{P}_{\text{pred}} \cap \mathcal{P}_{\text{gt}}|}{|\mathcal{P}_{\text{gt}}|}$$

where $\mathcal{P}_{\text{gt}}$ is the set of ground-truth predicates relevant to task completion (e.g., goal achievement, action pre-

conditions), and $\mathcal{P}_{\text{pred}}$ is the corresponding predicted set.

- **Symbolic Hallucination Rate (SymHalluc)**: The percentage of predicted task-relevant predicates that do not appear in the ground-truth symbolic world:

$$\text{SymHalluc} = \frac{|\mathcal{P}_{\text{pred}} \setminus \mathcal{P}_{\text{gt}}|}{|\mathcal{P}_{\text{pred}}|}$$

This measures the extent to which the model generates incorrect symbolic information related to the task, including hallucinated object states or spatial relations.

## C. Symbolic Execution Simulator Details

Figure 7 provides a qualitative example of the Symbolic Execution Simulator (SESim). Starting from an initial symbolic world and plan, SESim first performs symbolic validation to detect logical inconsistencies and then executes the plan in the simulator to obtain grounded feedback, which together are used to refine action plan and evolve the symbolic world.

### C.1. Symbolic Validation

The symbolic validation is implemented by wrapping an off-the-shelf PDDL validator. We use a standard VAL-style PDDL validator in all experiments. Given a symbolic world represented as a PDDL domain file and a problem file, together with a PDDL plan, the validator checks whether the plan is logically consistent with the current symbolic world. Concretely, the validator expands the plan step by step and verifies that, for each action: (i) all preconditions are satisfied in the current symbolic state, (ii) all referenced objects are defined and correctly typed, and (iii) the final state satisfies the goal conditions specified in the problem file.

If any of these checks fail, the validator returns a structured error trace containing unsatisfied preconditions, undefined or misspecified objects, and unmet goal predicates. We parse this trace into the symbolic validation feedback, which includes: (1) the index of the failing action, (2) the missing or violated predicates, and (3) the subset of goals that remain unsatisfied.

### C.2. Real Execution

In the real execution, the same PDDL plan is executed step-by-step in the simulator environment. Because the PDDL actions are defined at a high level with explicit preconditions and effects, we design the symbolic domain such that each PDDL action corresponds closely to a primitive executable action in the simulator. This allows us to directly map each action to an environment action and apply it to the current simulated state.

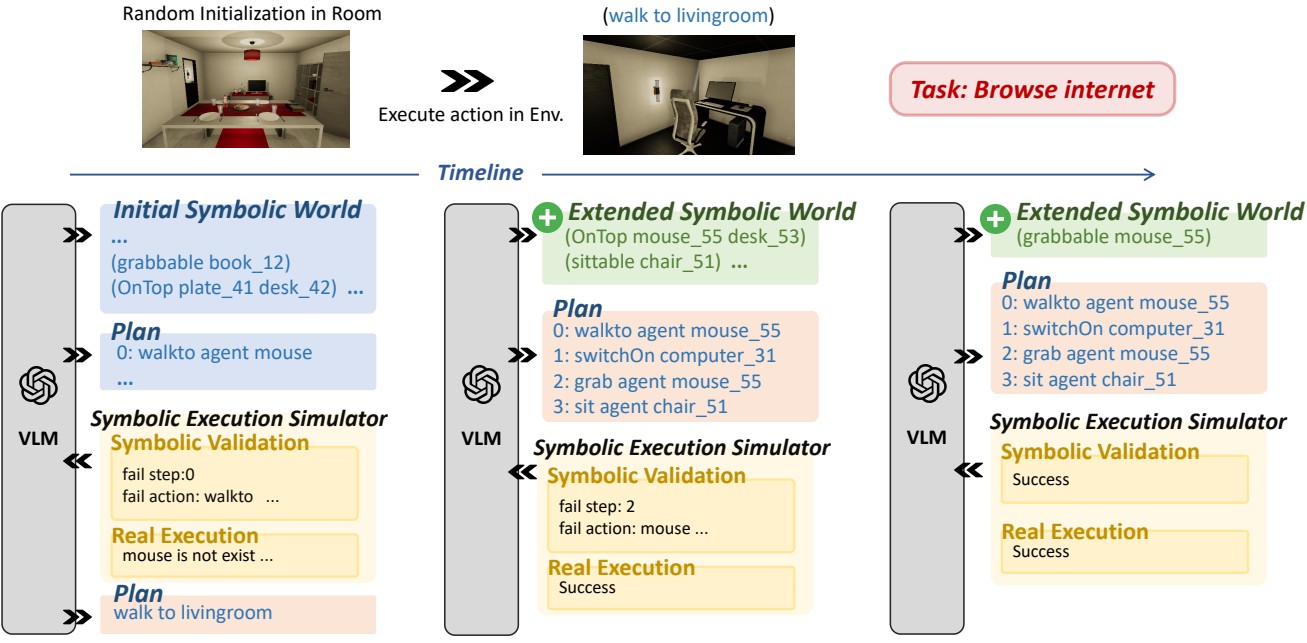

*Figure 7.* Example of how the Symbolic Execution Simulator (SESim) jointly uses symbolic validation and real execution feedback to refine both the action plan and the symbolic world.

For *VirtualHome*, we align PDDL actions with the built-in program-based API (e.g., `Walk`, `Grab`, `SwitchOn`), and execute them sequentially in the environment. The simulator reports whether an action succeeds, fails because the target object cannot be found, or fails because the action is not applicable in the current state (e.g., trying to `Open` an already open container). We log the pre- and post-execution observations and convert them into real execution feedback $\mathcal{F}_{\text{real}}$, which captures observed state changes, failed manipulations, and missing affordances.

For *ALFRED* tasks, we execute the plan using the ALF-World interface, adopting its discrete action format (e.g., `Goto`, `TakeObject`, `OpenObject`, `MoveObject`). The PDDL domain is constructed such that each high-level PDDL action has a one-to-one mapping to an *ALFWorld* action. During execution, *ALFWorld* returns success or failure signals together with updated observations.

Overall, the real execution stage provides grounded feedback that complements symbolic validation: even when a plan passes the PDDL validator, execution failures in the simulator reveal mismatches between the symbolic world and the embodied environment.

## D. Self-Adaptive Symbolic Memory Details

### D.1. When SASMem Is Queried

SASMem is queried *only when* SESim produces a structured failure signal, i.e., either (i) symbolic validation fails,

yielding $\mathcal{F}_{\text{symbol}}$, or (ii) symbolic validation passes but real execution fails (or reveals unexpected postconditions), yielding $\mathcal{F}_{\text{real}}$. Intuitively, SASMem is not used as a generic prompt augmentation; instead, it is activated by *localized* failure evidence so that the returned information is directly actionable for plan/world refinement.

**Query construction from structured feedback.** Given a symbolic validation failure, SESim extracts a compact query key from the failing action and the validator-reported error:

$$q_{\text{symbol}} = \langle \textit{fail\_action, error\_type, error\_content} \rangle.$$

Given a real execution failure or mismatch, SESim extracts an execution query key:

$$q_{\text{real}} = \langle \textit{fail\_content} \rangle$$

where *fail_content* summarizes the grounded failure evidence near the failure point (e.g., affordance mismatch, unexpected state change, or interaction refusal). SESim forwards these keys to a dedicated *MemoryManager*, which retrieves a small set of predicate-level *patches/hints* and returns them as $\mathcal{F}_{\text{SASMem}}$ for the next refinement call.

### D.2. What Is Returned

SASMem maintains two complementary stores: *action memory* and *world memory*. The MemoryManager returns only a minimal set of high-density hints rather than raw trajectories.

**Action reasoning memory.** Action memory is indexed by symbolic failure signatures and is retrieved primarily using $q_{symbol}$. For the most compatible entries (matched by the failing action and error type, and optionally refined by error content), each entry provides: (i) a short list of *failure-associated predicates* (the top predicates most frequently implicated by this failure signature), which serves as a hint for which symbolic facts are likely missing or inconsistent in the current world model; and (ii) a single *successful prerequisite action pattern* (top-1), describing the most common local prerequisite step that precedes this failing action in successful plans, which guides plan repair by suggesting which prerequisite should be ensured or inserted before the failing action.

**World modeling memory.** World memory is indexed by grounded mismatch evidence and is retrieved using $q_{real}$ jointly with the local failing action context. Each retrieved entry is predicate-aligned and provides: (i) a small set of predicate names with lightweight statistics indicating their tendency to be *missing* versus *mis-specified* under real interaction evidence (e.g., counts that accumulate missing-evidence vs error-evidence), which biases the VLM toward either adding a missing predicate instance or revising an incorrect one; and (ii) a single *probing action suggestion* (top-1) that most frequently surfaces interaction evidence for the predicate, enabling targeted exploration to confirm or refute the suspected symbolic gap.

### D.3. Write-gating with Credit Assignment

SASMem updates are governed by a write-gating mechanism that performs *credit assignment* based on whether an injected hint actually improves the subsequent SESim outcome. Concretely, suppose a SESim iteration first fails at step $t$ with

$$\langle \textit{fail\_step} = t, \textit{fail\_action} = a_t, \textit{error\_type} = e \rangle,$$

and the MemoryManager injects an action-memory entry associated with $(a_t, e)$. After refinement, SESim re-validates/re-executes and compares the next outcome against the original failure signature. The injected entry is treated as *effective* and its support statistic is increased if either: (i) the same error type $e$ no longer occurs at the same failing action $a_t$ (i.e., the original failure pattern is resolved), or (ii) the first failure moves to a strictly later step $t' > t$ (i.e., progress is made even if the task is not fully solved yet). Otherwise, if SESim still fails on the same action with the same symbolic error type, the entry is down-weighted via a symbolic-failure.

## E. PDDL Domain File

For completeness and reproducibility, we include the full PDDL domain definitions used in our experiments for both *VirtualHome* and *ALFRED*. These domains specify the types, predicates, and actions used for embodied planning in each environment.

### E.1. VirtualHome Domain

```
1  (define (domain virtualHome)
2   (:requirements :adl :strips :typing )
3
4   (:types
5      agent
6      room
7      object
8   )
9
10  (:predicates
11     (sitting_state ?a - agent); ?a is
            seated.
12     (lying_state ?a - agent); ?a is
            lying down.
13     (sleeping_state ?a - agent) ; ?a is
            asleep.
14     (clean_state ?obj - object) ; ?obj
            is clean
15     (turnon_state ?obj - object) ; ?obj
            is turned on.
16     (open_state ?obj - object) ; ?obj is
            open
17
18     (grabbable ?obj - object) ; ?obj can
            be picked up.
19     (sittable ?obj - object) ; ?obj can
            be sat on.
20     (eatable ?obj - object) ; ?obj can
            be eaten.
21     (readable ?obj - object) ; ?obj can
            be read.
22     (drinkable ?obj - object) ; ?obj can
            be drunk.
23     (switchable ?obj - object) ; ?obj
            can be switched on/off.
24     (openable ?obj - object) ; ?obj can
            be opened.
25
26     (agentAtRoom ?a - agent ?r - room) ;
             ?a is in room ?r
27     (agentAtObject ?a - agent ?obj -
            object); ?a is at ?obj
28     (holdsRight ?a - agent ?obj - object
            ) ; ?a holds ?obj in right hand
29     (holdsLeft ?a - agent ?obj - object)
             ; ?a holds ?obj in left hand
30     (isHoldingLeft ?a - agent) ; ?a
            holds an object in left hand
31     (isHoldingRight ?a - agent) ; ?a
            holds an object in right hand
32     (onTop ?obj1 - object ?obj2 - object
            ) ; ?obj1 is on top of ?obj2
33     (insideObject ?obj1 - object ?obj2 -
            object); ?obj1 is inside ?obj2
34     (objAtRoom ?obj - object ?r - room);
             ?obj is in room ?r
35  )
36
```

```
37  (:action walkToRoom
38    :parameters (?a - agent ?from - room
          ?to - room)
39    :precondition (and
40    (not(sitting_state ?a))
41    (agentAtRoom ?a ?from))
42    :effect (and
43    (agentAtRoom ?a ?to)
44    (not(agentAtRoom ?a ?from)))
45  )
46
47  (:action walkToObject
48    :parameters (?a - agent ?obj -
          object ?r - room)
49    :precondition (and
50    (not (sitting_state ?a))
51    (objAtRoom ?obj ?r)
52    (agentAtRoom ?a ?r))
53    :effect (agentAtObject ?a ?obj)
54  )
55
56  (:action runToRoom
57    :parameters (?a - agent ?from - room
          ?to - room)
58    :precondition (and
59    (not(sitting_state ?a))
60    (agentAtRoom ?a ?from))
61    :effect (and
62    (agentAtRoom ?a ?to)
63    (not(agentAtRoom ?a ?from)))
64  )
65
66  (:action runToObject
67    :parameters (?a - agent ?obj -
          object ?r - room)
68    :precondition (and
69    (not (sitting_state ?a))
70    (objAtRoom ?obj ?r)
71    (agentAtRoom ?a ?r))
72    :effect (agentAtObject ?a ?obj)
73  )
74
75  (:action sit
76    :parameters (?a - agent ?obj -
          object)
77    :precondition (and
78    (not (sitting_state ?a))
79    (agentAtObject ?a ?obj)
80    (sittable ?obj)
81    )
82    :effect (sitting_state ?a)
83  )
84
85  (:action standUp
86    :parameters (?a - agent)
87    :precondition (sitting_state ?a)
88    :effect (not (sitting_state ?a))
89  )
90
91  (:action grab
92    :parameters (?a - agent ?obj -
          object)
93    :precondition (and
94    (grabbable ?obj)
95    (agentAtObject ?a ?obj)
```

```
96      (or (not ( isHoldingLeft ?a)) (not (
            isHoldingRight ?a)))
97      )
98      :effect (and
99      (when (not(isHoldingRight ?a))
100       (and(holdsRight ?a ?obj) (
              isHoldingRight ?a)))
101     (when (and(isHoldingRight ?a) (not(
            isHoldingLeft ?a)))
102       (and(holdsLeft ?a ?obj) (
              isHoldingLeft ?a)))
103     )
104 )
105
106 (:action agent_open_object
107     :parameters (?a - agent ?obj -
            object)
108     :precondition (and
109     (openable ?obj)
110     (not(open_state ?obj))
111     (agentAtObject ?a ?obj)
112     (or (not (isHoldingRight ?a)) (not (
            isHoldingLeft ?a)))
113     )
114     :effect (open_state ?obj)
115 )
116
117 (:action agent_close_object
118     :parameters (?a - agent ?obj -
            object)
119     :precondition (and
120     (openable ?obj)
121     (open_state ?obj)
122     (agentAtObject ?a ?obj)
123     (or (not (isHoldingRight ?a)) (not (
            isHoldingLeft ?a)))
124     )
125     :effect (not(open_state ?obj))
126 )
127
128 (:action putOn
129     :parameters (?a - agent ?obj1 -
            object ?obj2 - object)
130     :precondition (and
131     (or (holdsRight ?a ?obj1) (holdsLeft
            ?a ?obj1))
132     (agentAtObject ?a ?obj2)
133     )
134     :effect (and
135     (when(holdsRight ?a ?obj1)
136       (and(not(holdsRight ?a ?obj1))(
              not(isHoldingRight ?a))))
137     (when(not(holdsRight ?a ?obj1))
138       (and(not(holdsLeft ?a ?obj1))(
              not( isHoldingLeft ?a))))
139     (onTop ?obj1 ?obj2)
140     )
141 )
142
143 (:action putIn
144     :parameters (?a - agent ?obj1 -
            object ?obj2 - object)
145     :precondition (and
146     (or (holdsRight ?a ?obj1) (holdsLeft
            ?a ?obj1))
```

```
147        (agentAtObject ?a ?obj2)
148        (open_state ?obj2)
149        )
150        :effect (and
151        (when(holdsRight ?a ?obj1)
152            (and(not(holdsRight ?a ?obj1))(
                    not(isHoldingRight ?a))))
153        (when(not(holdsRight ?a ?obj1))
154            (and(not(holdsLeft ?a ?obj1))(
                    not( isHoldingLeft ?a))))
155        (insideObject ?obj1 ?obj2)
156        )
157    )
158
159    (:action switchOn
160        :parameters (?a – agent ?obj –
                object)
161        :precondition (and
162        (switchable ?obj)
163        (not(turnon_state ?obj))
164        (agentAtObject ?a ?obj)
165        )
166        :effect (turnon_state ?obj)
167    )
168
169    (:action switchOff
170        :parameters (?a – agent ?obj –
                object)
171        :precondition (and
172        (switchable ?obj)
173        (turnon_state ?obj)
174        (agentAtObject ?a ?obj)
175        )
176        :effect (not(turnon_state ?obj))
177    )
178
179    (:action drink
180        :parameters (?a – agent ?obj –
                object)
181        :precondition (and
182        (drinkable ?obj)
183        (agentAtObject ?a ?obj)
184        )
185        :effect (and)
186    )
187
188    (:action touch
189        :parameters (?a – agent ?obj –
                object)
190        :precondition (agentAtObject ?a ?obj
                )
191        :effect (and)
192    )
193
194    (:action lookAt
195        :parameters (?a – agent ?obj –
                object)
196        :precondition (agentAtObject ?a ?obj
                )
197        :effect (and)
198    )
199
200    (:action wipe
201        :parameters (?a – agent ?obj –
                object)
```

```
202        :precondition (and
203        (or (holdsRight ?a ?obj) (holdsLeft
                ?a ?obj))
204        (agentAtObject ?a ?obj)
205        )
206        :effect (clean_state ?obj)
207    )
208
209    (:action wash
210        :parameters (?a – agent ?obj –
                object)
211        :precondition (and
212        (or (holdsRight ?a ?obj) (holdsLeft
                ?a ?obj))
213        (agentAtObject ?a ?obj)
214        )
215        :effect (clean_state ?obj)
216    )
217
218    (:action read
219        :parameters (?a – agent ?obj –
                object)
220        :precondition (and
221        (or (holdsRight ?a ?obj) (holdsLeft
                ?a ?obj))
222        (agentAtObject ?a ?obj)
223        (readable ?obj)
224        )
225        :effect (and)
226    )
227
228    (:action eat
229        :parameters (?a – agent ?obj –
                object)
230        :precondition (and
231        (or (holdsRight ?a ?obj) (holdsLeft
                ?a ?obj))
232        (agentAtObject ?a ?obj)
233        (eatable ?obj)
234        )
235        :effect (and)
236    )
237
238    (:action sleep
239        :parameters (?a – agent)
240        :precondition (or (sitting_state ?a)
                (lying_state ?a))
241        :effect (sleeping_state ?a)
242    )
243
244    (:action wakeup
245        :parameters (?a – agent)
246        :precondition (sleeping_state ?a)
247        :effect (not(sleeping_state ?a))
248    )
249 )
```

## E.2. ALFRED Domain

```
1 (define (domain alfred)
2 (:requirements
3    :adl
4    :typing
5 )
```

```
 6  (:types
 7   agent
 8   receptacle
 9   object
10   )
11
12  (:predicates
13     (agentAtReceptacle ?a - agent ?r -
           receptacle)              ; true if
           the agent is at the receptacle
14     (objectAtReceptacle ?o - object ?r -
            receptacle)             ; true
           if the object is at the
           receptacle
15
16     (openable ?r - receptacle)
                                        ;
           true if a receptacle is openable
17     (cleanable ?o - object)

           ; true if the object can be
           placed in a sink
18     (heatable ?o - object)

           ; true if the object can be
           heated up in a microwave
19     (coolable ?o - object)

           ; true if the object can be
           cooled in the fridge
20     (pickupable ?o - object)

           ; true if the object can be
           picked up
21     (toggleable ?o - object)

           ; true if the object can be
           turned on/off
22     (cleanTool ?r - receptacle)
23     (heatTool ?r - receptacle)
24     (coolTool ?r - receptacle)
25
26     (holds ?a - agent ?o - object)
                                        ;
           object ?o is held by agent ?a
27     (holdsAny ?a - agent)

           ; agent ?a holds an object
28     (opened ?r - receptacle)

           ; true if a receptacle is opened
29     (isClean ?o - object)

           ; true if the object has been
           clean in sink
30     (isHot ?o - object)

           ; true if the object has been
           heated up
31     (isCool ?o - object)

           ; true if the object has been
           cooled
32     (isOn ?o - object)
           ; true if the object is on
33  )
34  (:action GotoReceptacle
35     :parameters (?a - agent ?from_recep
           - receptacle ?to_recep -
           receptacle )
36     :precondition (and
37         (agentAtReceptacle ?a ?
               from_recep)
38         (not (agentAtReceptacle ?a ?
               to_recep))
39         )
40     :effect (and
41         (not (agentAtReceptacle ?a ?
               from_recep))
42         (agentAtReceptacle ?a ?to_recep)
43         )
44  )
45
46  (:action OpenReceptacle
47     :parameters (?a - agent ?r -
           receptacle)
48     :precondition (and
49         (openable ?r)
50         (agentAtReceptacle ?a ?r)
51         (not (opened ?r))
52         )
53     :effect (and
54         (opened ?r)
55         )
56  )
57  (:action CloseObject
58     :parameters (?a - agent ?r -
           receptacle)
59     :precondition (and
60         (openable ?r)
61         (agentAtReceptacle ?a ?r)
62         (opened ?r)
63         )
64     :effect (and
65         (not (opened ?r))
66         )
67  )
68
69  (:action TakeObject
70     :parameters (?a - agent ?o - object
           ?r - receptacle)
71     :precondition
72         (and
73             (pickupable ?o)
74             (agentAtReceptacle ?a ?r)
75             (objectAtReceptacle ?o ?r)
76             (not (holdsAny ?a))  ; agent
                   's hands are empty.
77             (or (not (openable ?r)) (
                   opened ?r))
78         )
79     :effect
80         (and
81             (not (objectAtReceptacle ?o
                   ?r))
82             (holds ?a ?o)
83             (holdsAny ?a)
84         )
85  )
```

```
86
87   (:action MoveObject
88       :parameters (?a – agent ?o – object
             ?r – receptacle )
89       :precondition (and
90               (holds ?a ?o)
91               (agentAtReceptacle ?a ?r)
92               (or (not (openable ?r)) (
                   opened ?r))
93               )
94       :effect (and
95               (objectAtReceptacle ?o ?r)
96               (not (holds ?a ?o))
97               (not (holdsAny ?a))
98               )
99   )
100
101  (:action CleanObject
102      :parameters (?a – agent ?o – object
             ?r – receptacle)
103      :precondition (and
104              (cleanable ?o)
105              (cleanTool ?r)
106              (agentAtReceptacle ?a ?r)
107              (holds ?a ?o)
108              )
109      :effect (and
110                (isClean ?o)
111              )
112  )
113
114  (:action HeatObject
115      :parameters (?a – agent ?o – object
             ?r – receptacle)
116      :precondition (and
117              (heatable ?o)
118              (heatTool ?r)
119              (agentAtReceptacle ?a ?r)
120              (holds ?a ?o)
121              )
122      :effect (and
123                (isHot ?o)
124                (not (isCool ?o))
125              )
126  )
127
128  (:action CoolObject
129      :parameters (?a – agent ?o – object
             ?r – receptacle )
130      :precondition (and
131              (coolable ?o)
132              (coolTool ?r)
133              (agentAtReceptacle ?a ?r)
134              (holds ?a ?o)
135              )
136      :effect (and
137                (isCool ?o)
138                (not (isHot ?o))
139              )
140  )
141
142  (:action ToggleObject
143      :parameters (?a – agent ?o – object
             ?r – receptacle)
144      :precondition (and
```

```
145              (toggleable ?o)
146              (agentAtReceptacle ?a ?r)
147              (objectAtReceptacle ?o ?r)
148              )
149      :effect (and
150              (when (isOn ?o)
151                  (not (isOn ?o)))
152              (when (not (isOn ?o))
153                  (isOn ?o))
154              )
155  )
156
157  )
```

