# OpenReview forum: "SCOPE: Evolving Symbolic World for Planning in Open-Ended Environments"
_ICML.cc/2026/Conference — ICML 2026 regular_

### Official Review · Reviewer_LDrz · 2026-03-06

**Soundness:** 3
**Presentation:** 3
**Significance:** 3
**Originality:** 3
**Overall Recommendation:** 4
**Confidence:** 4

**Summary:**

This work proposes SCOPE, which is an adaptive symbolic planning framework for open-ended environments. SCOPE integrates action plan refinement and symbolic world evolution into a single closed-loop system. Particularly, SESim updates plans and the symbolic world by combining symbolic verification with feedback from actual execution, while SASMem distills this feedback into symbolic knowledge that can be reused across tasks. Extended simulation on VirtualHome and ALFRED demonstrates the superior performance of the SCOPE.

**Compliance With Llm Reviewing Policy:**

Affirmed.

**Key Questions For Authors:**

Please refer to the weaknesses. Moreover,
1. Does SASMem's write policy remain stable under real robot or noisy execution feedback?

**Limitations:**

Yes

**Strengths And Weaknesses:**

Strengths:
1. The static properties of the symbolic world and the inability to distill experience into reusable symbolic knowledge represent two major challenges in open-ended environments. This paper formulates these two problems and proposes two corresponding modules for solutions.
2. The proposed SCOPE not only enhances planning but also elevates the quality of the symbolic world. Table 2 shows that under the open-ended setting, SCOPE also significantly outperforms variants that omit SESim/SASMem on both ClassicalSR and SymRecall.
3. Extended simulation on VirtualHome and ALFRED demonstrates significant performance of the SCOPE compared to several baselines.
Weaknesses:
1. The novelty of the paper is limited. In particular, compared to ISR-LLM, NESYC, and memory-augmented embodied agents, SCOPE  does not represent concept or theoretical innovation but rather appears to be a combination of existing techniques, such as symbolic feedback, real-world execution feedback, and experiential memory.
2. The perfromance rely on simulator-friendly feedback, which limits the application in the real world.
3. The existing phase-based novelty task pipeline emphasizes knowledge retention across tasks but does not account for uncontrollable changes in open-world environments. In other words, the current setup resembles a well-designed phased novelty test rather than a true open-world deployment.

---

> ### Author Rebuttal · Authors · 2026-03-31
>
> **W1: Novelty of the contribution**
>
> **A1**: We appreciate your concern regarding novelty. We would like to clarify that compared with prior work, **SCOPE’s core insight is that under an incomplete symbolic world, the key capability is not planning over a fixed symbolic world, but diagnosing and repairing plan–world inconsistencies**, with symbolic validator serving as a structured diagnostic interface.
> - SESim turns failure into two complementary signals: symbolic validation provides interpretable and localized evidence of plan-side reasoning errors under the current symbolic world, while real execution provides grounded evidence of world-side grounding errors caused by incomplete symbolic modeling. **These two signals offer complementary explanations for failure**.
> - SASMem is not merely a memory module, but **a mechanism for error disentanglement and cross-task credit assignment**: it separates experience into predicate-level action-reasoning knowledge and world-modeling knowledge, and applies write-gating only when retrieved knowledge resolves the same failure pattern or delays the first failure step.
>
> **W2: Feedback dependency**
>
> **A2**: Thanks for raising this important concern. To examine it, we evaluated SCOPE under two harder settings that are closer to real-world deployment: (1) success/failure only, without any failure content, and (2) a VLM-based monitor that predicts success/failure from pre-/post-action visual observations (in our reply to Reviewer **NNJP's W1**). Empirically, SCOPE remains **effective under binary-only feedback** and outperform the baselines even under the noisier VLM-monitor setting.
>
> **W3: Open-world setup**
>
> **A3**: Thank you for this important comment. We agree that in a real open-world deployment, novelty may occur randomly and uncontrollably rather than only in phase-wise transitions. To address this concern, we additionally evaluate three randomized variants: (1) **Random-Env**, where environment changes occur at independently randomized times; (2) **Random-Afford**, where task/action-affordance novelty occurs at independently randomized times; and (3) **Random-Mixed**, which combines both.
> - As the table below shows, SCOPE remains strong under randomized and non-phase-aligned novelty, suggesting that its gains are not tied to phase boundaries and that it can handle unpredictable environment/task changes robustly.
> - Our original phase-based setup is a controlled diagnostic protocol for **clearly measuring adaptation under different novelty types**, rather than assuming real-world changes are phase-structured. We will add the results and clarify this motivation in the final version.
>
> |Method|Phased|Random-Env SR|Random-Afford SR|Random-Mixed SR|
> |-|-|-|-|-|
> |ISR-LLM|55.2|54.9|55.3|54.7|
> |NESYC|51.0|49.3|69.5|48.8|
> |SCOPE|83.1|82.8|83.3|81.9|
>
> **Q1: SASMem's write policy**
>
> **A4**: Thank you for the important question. We would like to clarify that SASMem’s stability does not come from assuming clean feedback. Specifically, **SASMem stores predicate-level distilled action/world knowledge rather than raw traces, and it is activated only by localized failure evidence**.
>
> To evaluate this, we add a memory contamination experiment (in our reply to Reviewer **2Rg5's Q3**) and observe that retrieval cleanliness remains relatively high compared with the uncontaminated memory, while SR degrades only moderately as contamination increases, indicating that **the write policy can, to some extent, suppress error accumulation rather than amplify it**.

---

> > ### Author Rebuttal · Reviewer_LDrz · 2026-04-01
> >
> > Thank you for your response. There is no further question.

---

### Official Review · Reviewer_NNJP · 2026-03-10

**Soundness:** 3
**Presentation:** 3
**Significance:** 2
**Originality:** 3
**Overall Recommendation:** 4
**Confidence:** 4

**Summary:**

SCOPE is a framework designed to help AI agents plan and complete tasks in environments that are constantly changing or not fully known at the start. Most current systems create a single and static map of the world and get stuck if something is different than expected. SCOPE fixes this by creating a symbolic world model that evolves.The system uses two main parts. SESim that checks if a plan makes sense logically and then tries it out. If the plan fails, it uses that feedback to update its internal environment model. SASMem that stores what the agent has learned from past mistakes so it can solve future tasks faster without repeating the same errors, basically a memory bank. By constantly updating its understanding of the world and remembering past experiences, the agent becomes much better at handling long, complicated tasks in unpredictable settings, in the virtual environments studied in this work.

**Compliance With Llm Reviewing Policy:**

Affirmed.

**Final Justification:**

I stand by my initial rating after the authors' reply.

**Key Questions For Authors:**

How would the authors adapt SCOPE to handle messy, continuous sensory data of a real robot where failure isn't always clearly categorized?

How can the method be applied and adopted to real-world environment that is might contain object states beyond what the current PDDL can describe?

To what extent is the success of SCOPE dependent on the specific phrasing of the prompts used for the VLM?

What is the algorithmic contribution of the work, apart from the intuitive motivation that updating the environment representation after changes and keeping track of interaction feedback in memory should help?

**Limitations:**

Yes.

**Strengths And Weaknesses:**

Strengths

Addressing Open-Ended Challenges: The paper tackles the difficult problem of open-ended environments where task-relevant objects and rules aren't pre-defined and can change over time. This is a significant hurdle for traditional planning models.

Intuitive and Systematic Solution: The authors propose a sensible approach to this complexity: maintaining a regularly updated symbolic representation of the environment and using a structured memory to keep track of knowledge gained through interaction.

Effective for Long-Horizon Tasks: The framework proves effective in scenarios requiring long sequences of interdependent actions. Results show significant improvements in task success rates in complex, multi-step environments like VirtualHome and ALFRED.

Weaknesses

Limited and Naturally Symbolic Environments: The evaluation is restricted to the VirtualHome and ALFRED simulators. These environments are inherently symbolic, as actions and states are already defined by program-based APIs and environments can be natively formulated as PDDL. It remains unknown if this method would be effective in a real-world robotics environment where sensor noise is high and feedback isn't neatly delivered as text-based error codes or when the environment contains many states not easily represented in a limited domain PDDL languages.

Vague Technical Details: Several implementation details are unclear. For instance, the paper does not explicitly state how frequently the environment representation is updated or what specific triggers prompt a full world evolution.

System Over-Crafting: The framework appears to be a highly crafted system specifically tuned for the provided simulators. Many technical solutions seem to involve simply formulating environment feedback into plain language and feeding them as prompts into a Vision-Language Model (VLM), which raises questions about the true depth of the contribution being performed.

---

> ### Author Rebuttal · Authors · 2026-03-31
>
> **W1: Environment setting**
>
> **A1**: Thank you for the valuable question. To probe this concern, we further modify the execution-feedback channel to better approximate real-world conditions under simulator. Specifically, we consider three settings: (1) the original setup in our paper; (2) a harder variant where we keep **only binary success/failure** feedback; and (3) a **VLM-based monitor**, which takes as input the action arguments with visual observations before and after execution, and predicts only success/failure. The third setting is intended to better mimic real-world robotics, where action outcomes are inferred from perception rather than returned as structured textual error codes; correspondingly, setting (2) can be viewed as an approximate upper bound for setting (3). Empirically, SCOPE still performs well when only binary simulator feedback is available.
>
> |Method|Execution feedback source|Returned|Static SR|Dynamic SR|Open-ended SR|
> |-|-|-|-|-|-|
> |ISR-LLM|simulator|success/failure + failure content|61.4|58.9|54.7|
> |NESYC|simulator|success/failure + failure content|88.5|72.8|53.9|
> |SCOPE|simulator|success/failure + failure content|87.7|85.0|83.8|
> |ISR-LLM|simulator|success/failure only|53.1|46.6|43.2|
> |NESYC|simulator|success/failure only|76.9|60.2|41.1|
> |SCOPE|simulator|success/failure only|77.1|73.5|70.0|
> |ISR-LLM|VLM monitor|success/failure only|45.8|40.7|36.6|
> |NESYC|VLM monitor|success/failure only|68.0|63.1|38.8|
> |SCOPE|VLM monitor|success/failure only|70.2|68.9|67.2|
>
> **W2: Technical Details**
>
> **A2**: Thank you for pointing this out. In our implementation, the symbolic world is not updated at a fixed frequency. Instead, **world evolution is event-driven**. We first construct an initial symbolic world from visual observations and the task description by VLM, and updates are triggered in two cases: (1) when the VLM explicitly chooses to explore; (2) when SESim reports issues such as undefined objects or incomplete goal satisfaction. Importantly, these **updates are localized plan/world refinements around the detected failure evidence**. We will clarify this point in the revised version.
>
> **W3: Framework design**
>
> **A3**: Thank you for this important concern. We would like to clarify that SCOPE's key contribution is a structured diagnosis-and-repair framework for planning under incomplete symbolic worlds.
> - SESim separates failure into two complementary sources: symbolic validation provides interpretable evidence of plan-side inconsistencies under the current symbolic world, while real execution provides grounded evidence of world-side mismatches. The role of **SESim is to identify where the inconsistency lies and what type of repair is needed**.
> - SASMem algorithmically transforms feedback into **high-information-density, symbol-friendly action/world knowledge**, enabling cross-task reuse of repair patterns.
>
> **Q1: Continuous sensory feedback**
>
> **A4**: Thank you for the important question. We further evaluate SCOPE under two harder settings (in our reply to Reviewer **NNJP's W1**). These results suggest that **SCOPE does not fundamentally rely on neatly categorized textual error codes**.
> - For real robots, we would replace simulator failure content with a perception-based monitor over visual observations and robot signals (e.g., proprioception/gripper status), producing soft success/failureure estimates.
>
> **Q2: Beyond fixed PDDL**
>
> **A5**: Thank you for this important question. **SCOPE can also support more open-ended growth of the symbolic representation** (in our reply to Reviewer **p4ht’s W1**).
> - For real-world deployment, when execution evidence indicates that a previously unmodeled state is needed for planning, the VLM can propose new predicates.
>
> **Q3: Prompt sensitivity and robustness**
>
> **A6**: Thank you for this question. To assess this, we additionally evaluated both ISR-LLM and SCOPE under three prompt variants using two backbones. The results indicate that SCOPE is robust to moderate prompt variations rather than relying on a specific phrasing. We will clarify this point in the revised version.
>
> |Method|model|original prompt SR|concise paraphrase SR|reordered instruction SR|Prompt-Std SR|
> |-|-|-|-|-|-|
> |ISR-LLM|Qwen3-VL-8B|55.2|45.9|49.2|3.85|
> |ISR-LLM|GPT-4o|74.5|67.4|71.9|2.94|
> |SCOPE|Qwen3-VL-8B|64.9|58.1|63.2|2.89|
> |SCOPE|GPT-4o|84.5|81.4|84.6|1.49|
>
> **Q4: Algorithmic novelty**
>
> **A7**: Thank you for this important question. Our algorithmic contribution is **an explicit diagnosis-and-repair framework for planning under incomplete symbolic worlds**. SESim uses complementary signals from symbolic validation and real execution to determine whether the VLM should refine the current plan, evolve the symbolic world, or update both. SASMem provides a predicate-level memory mechanism with write-gating for credit assignment and noise suppression.

---

> > ### Author Rebuttal · Reviewer_NNJP · 2026-04-01
> >
> > The authors address some of my concerns. But I believe the framework would be less effective in a real-world environment with complex visual input. My rating on the quality of the paper does not change.

---

> > > ### Author Response · Authors · 2026-04-01
> > >
> > > Thank you again for your careful reading and thoughtful feedback. We appreciate that some of our clarifications were helpful. We also understand your concern about effectiveness in real-world environments with complex visual input. This is an important consideration for assessing practical deployment. In the current paper, our goal is to study the proposed framework under open-ended settings, so as to isolate whether it can generate long-horizon and grounded plans . We will clarify this empirical scope more explicitly in the revision.

---

### Official Review · Reviewer_2Rg5 · 2026-03-13

**Soundness:** 2
**Presentation:** 3
**Significance:** 2
**Originality:** 3
**Overall Recommendation:** 4
**Confidence:** 3

**Summary:**

This paper proposes SCOPE, a neuro-symbolic framework for embodied agent planning. The framework combines a vision–language model (VLM) with a symbolic simulator. Given an input, the VLM generates a symbolic plan, which is then executed within the symbolic simulator to iteratively verify its correctness through feedback. In addition, the framework includes a symbolic memory module that maintains feedback from previous executions and uses it as a prior for future inference. Experiments on ALFRED and VirtualHome demonstrate that the proposed method outperforms several baseline approaches, especially under settings that introduce new settings.

**Compliance With Llm Reviewing Policy:**

Affirmed.

**Final Justification:**

The additional baselines make the paper more solid, and I will raise my score to 4. However, I still find the “world modeling” claim overstated, as the method mainly refines/debugs a predefined symbolic space rather than genuinely world modeling.

**Key Questions For Authors:**

Questions
1. Can SCOPE be combined with a classical planner for a new baseline? More specifically, could the simulator-refinement and evolving mechanism be integrated with a traditional planning pipeline?
2. Could cases like the one shown in Figure 6 be addressed using a replanning approach such as ReAct [1]?
3. How does SCOPE avoid error or hallucination accumulation? For example, if an execution error occurs in an early step and the feedback mistakenly assumes that the symbolic world needs to be updated, this could introduce an incorrect rule. How would such errors be corrected in later steps?
4. Since ALFRED and VirtualHome are virtual environments generated from symbolic rules, what modifications would be necessary for the proposed method to operate effectively in real-world environments?

[1] Shunyu Yao, Jeffrey Zhao, Dian Yu, Nan Du, Izhak Shafran, Karthik Narasimhan, and Yuan Cao. React: Synergizing reasoning and acting in language models. ICLR 2023.

**Limitations:**

yes

**Strengths And Weaknesses:**

Strengths
1. The proposed method effectively integrates neural and symbolic components for embodied agent planning, leveraging the advantages of symbolic reasoning.
2. Experiments on ALFRED and VirtualHome demonstrate that the proposed approach achieves clear improvements over baselines when continuously introducing new settings.
3. The paper provides extensive ablation studies to analyze the contribution of different components.

Weaknesses
1. The claim regarding world modeling appears somewhat overstated. Both ALFRED and VirtualHome are virtual environments designed by humans using relatively simple rules, which themselves originate from a limited set of symbolic rules. As a result, the evolution of the symbolic simulator in this framework can be regarded as detecting symbolic mismatches or debugging PDDL, rather than world modeling.
2. The baseline setup raises some concerns, as discussed in the questions below.

---

> ### Author Rebuttal · Authors · 2026-03-31
>
> **W1: World modeling claim**
>
> **A1**: Thanks for the important clarification. Our key point is that the symbolic world in SCOPE is not fixed: it is continuously expanded with new objects, state predicates, property predicates, and relational predicates through exploration and interaction. Such a view is consistent with prior work that treats world modeling as **building structured, relational, updatable, and retrievable world representations** for downstream reasoning and planning[1][2]. In this sense, **SCOPE evolves a symbolic world that is updatable, retrievable, and reusable over long horizons**. As the table below shows, the evolved symbolic world becomes substantially richer over time.
>
> [1]Yan Z, Li S, Wang Z, et al. Dynamic open-vocabulary 3d scene graphs for long-term language-guided mobile manipulation[J]. IEEE Robotics and Automation Letters, 2025.
> [2]Yang Y, Yang H, Zhou J, et al. 3D-mem: 3D scene memory for embodied exploration and reasoning[C]//Proceedings of the Computer Vision and Pattern Recognition Conference. 2025: 17294-17303.
>
> |type|Avg. Initial Count|Avg. Evolved Count|Increase(%)|
> |-|-|-|-|
> |object|24|51|+112.5|
> |state predicate|17|61|+258.8|
> |property predicate|10|32|+220.0|
> |relation predicate|52|174|+234.6|
>
> **Q1: Integration with classical planning**
>
> **A2**: Thank you very much for this valuable suggestion. We further supplemented the experiments by replacing the VLM planner in SCOPE with a classical planner. As shown in the table below:
> - Replacing the VLM with a classical planner slightly improves performance in the static setting, where the symbolic world is relatively complete.
> - However, in more complex environments with incomplete symbolic world generated from VLM, the **classical planner often cannot generate a plan and thus provides little guidance on what symbolic information is absent**.
> - In contrast, **the VLM planner can still propose exploratory actions**, which are then checked by the classical validator to **identify violated preconditions or conflict points**, providing actionable signals for refinement.
>
> |Method|Planner|Static SR|Dynamic SR|Open-ended SR|
> |-|-|-|-|-|
> |SCOPE|classical planner|87.9|74.7|56.7|
> |SCOPE|VLM|86.7|82.4|80.2|
>
> **Q2: ReAct baseline**
>
> **A3**: Thank you for the valuable question. Following your suggestion, we compare SCOPE with **ReAct**. As the table below shows, while generic replanning can help recover some local failures, SCOPE achieves stronger overall SR, indicating that the gain comes not only from replanning itself, but also from feedback-grounded symbolic diagnosis and symbolic-world update in open-ended environments. The comparison further suggests that **pure VLM-based replanning becomes increasingly limited as the environment grows more complex**.
>
> |Method|Static SR|Dynamic SR|Open-ended SR|
> |-|-|-|-|
> |ReAct|60.7|58.4|55.2|
> |SCOPE|86.7|82.4|80.2|
>
> **Q3: Robustness to error accumulation**
>
> **A4**: Thank you for this important question. Our symbolic world in SCOPE is updated continuously rather than fixed once, so an incorrect early update can be corrected if it later leads to planning-relevant errors. Moreover, SASMem uses a write-gating mechanism to prevent noisy memories from being persistently amplified.
>
> To make this explicit, we further supplemented a memory contamination study: we replace **a fraction ρ of SASMem memory mass** with incompatible predicate/action hints sampled from other keys, and evaluate retrieval robustness via **Action/World Clean@k**. Action/World Clean@k denotes the percentage of action-memory/world-memory queries for which the correct hint, defined by the uncontaminated memory for the same query key, still appears in the top-k retrieved results after corruption. Across contamination, while SR decreases only moderately, suggesting that **SCOPE could suppress, rather than amplify, error accumulation**.
>
> |Noise ratio 𝜌|Action Clean@1|Action Clean@3|World Clean@1|World Clean@3|SR|
> |-|-|-|-|-|-|
> |10%|100.0|95.6|100.0|93.2|80.3|
> |20%|99.2|94.5|99.4|92.9|78.4|
> |30%|98.9|94.1|97.9|92.6|78.0|
>
> **Q4: Real-world deployment**
>
> **A5**: Thank you for the valuable question. From the perspective of the SCOPE framework, the main modification needed for real-world deployment is to replace the simulator-provided execution feedback with a perception-based monitor, rather than to redesign the method itself. **SCOPE does not fundamentally rely on structured textual error messages**; it only requires an execution-feedback channel that reveals whether the current symbolic world matches the actual post-action outcome. In real-world robotics, this signal would be inferred from pre/post-action observations (and potentially other robot signals).
>
> To examine this, we further evaluated two harder settings(in our reply to Reviewer **NNJP's W1**). Empirically, SCOPE remains robust under binary-only feedback and outperforms baselines even under the noisier VLM-monitor setting.

---

> > ### Author Rebuttal · Reviewer_2Rg5 · 2026-04-02
> >
> > The additional baselines make the paper more solid, and I will raise my score to 4. However, I still find the “world modeling” claim overstated, as the method mainly refines/debugs a predefined symbolic space rather than genuinely world modeling.

---

### Official Review · Reviewer_p4ht · 2026-03-13

**Soundness:** 3
**Presentation:** 3
**Significance:** 2
**Originality:** 2
**Overall Recommendation:** 4
**Confidence:** 3

**Summary:**

This paper introduces SCOPE, a self-adaptive symbolic planning framework for embodied agents in open-ended environments.
SCOPE constructs a symbolic world represented in PDDL and iteratively refines both the symbolic world and the action plan through SESim. In addition, SASMem distills feedback into action reasoning knowledge and world modeling knowledge, enabling improved long-horizon planning. Experiments on ALFRED and VirtualHome show that SCOPE improves symbolic world completeness and robustness of planning under open-ended environments.

**Compliance With Llm Reviewing Policy:**

Affirmed.

**Final Justification:**

I appreciate the authors’ careful and detailed response. The concerns I raised regarding the weakness have been addressed. However, I remain cautious about such a core issue being resolved within such a short period of time.

Therefore, I will keep my original score.

**Key Questions For Authors:**

Please see the weaknesses.

**Limitations:**

yes

**Strengths And Weaknesses:**

### Strengths:
This paper proposes a closed-loop framework that leverages both symbolic validation and real execution feedback to evolve the symbolic world. It further introduces SASMem, enabling improved long-horizon planning, cross-task grounding, and adaptation across diverse environments. The experimental results on ALFRED and VirtualHome show that the proposed framework improves symbolic world completeness and planning robustness, including under environment perturbations. In addition, the ablation studies highlight the complementary contributions of SESim and SASMem to the overall performance.

### Weaknesses:
Although SCOPE is framed as “evolution,” what it actually performs is closer to refinement of the grounded symbolic state within a fixed PDDL schema, rather than open-ended growth of the symbolic representation, such as expansion of the predicate set or transition rules.

In Figure 6, the point that it is an inevitability of the evolving symbolic world does not seem convincing to me. The example appears to show that the initial plan simply made a semantically ignored action choice by VLMs, which was then corrected by symbolic validation.

So the example is more like a case of fixing a poor initial action selection rather than demonstrating that the symbolic world itself needed to evolve. It would be helpful to show a case where updating the world state directly changes downstream plan feasibility or enables a more rational and efficient actions [1, 2].

[1] Choi, Wonje, et al. "Nesyc: A neuro-symbolic continual learner for complex embodied tasks in open domains." ICLR 2025.

[2] Cornelio, Cristina, and Mohammed Diab. “Recover: A Neuro-Symbolic Framework for Failure Detection and Recovery.” IROS 2024.

Although SASMem decomposes experience into action-level and predicate-level entries in a way that improves transferability, its current design appears to remain closer to structured symbolic storage. A promising direction for further improvement would be to extend the memory itself to induce more generalized symbolic knowledge or abstracted concepts, thereby improving its intrinsic reusability.

---

> ### Author Rebuttal · Authors · 2026-03-31
>
> **W1: Open-endedness of symbolic evolution**
>
> **A1**: Thanks for your insightful observation. Our method is not inherently limited to refining the grounded symbolic state under a fixed schema. **SCOPE can also support more open-ended growth of the symbolic representation**. To verify this, we conducted additional experiments in the VirtualHome open-ended setting where schema-level growth is allowed during evolution. Specifically, we consider two forms of expansion: (1) **predicate set expansion**, where the VLM proposes new predicates to capture newly discovered object properties or relations; and (2) **transition rule expansion**, which in our setting is instantiated as adding new composed operators built from existing atomic actions.  The results are shown in the following table. SCOPE can realize open-ended growth of the symbolic representation beyond fixed-state refinement, while maintaining comparable planning performance.
>
> In the current submission we adopt a fixed PDDL schema in the main experiments because **it provides stronger interpretability for the final plans and enables more reliable symbolic validation**. In contrast, when the schema itself is expanded autonomously by the VLM, the resulting symbolic validation can become less reliable. We will clarify it more explicitly in the revision.
>
> |Method|predicate set change?|transition rules change?|SR|GC|predicate set count|transition rules count|
> |-|-|-|-|-|-|-|
> |SCOPE|No|No|80.1|83.4|22|22|
> |SCOPE|Yes|No|78.2|79.5|25|22|
> |SCOPE|No|Yes|76.9|77.5|22|26|
> |SCOPE|Yes|Yes|76.5|78.4|24|25|
>
> **W2: Interpretation of Figure 6**
>
> **A2**: Thanks for pointing out this ambiguity. We agree that the current presentation of Figure 6 may be somewhat misleading to readers. In this example, after the Symbolic Execution Simulator identifies the invalid action, the VLM **interacts with the environment to recover the missing predicates** related to faucet and barsoap before refining the invalid action. In the revised version, we will explicitly clarify this process below the figure by adding: “Update symbolic world: (switchable faucet), (grabbable barsoap), (not(turnon_state faucet))”. This clarification better reflects that the refinement is driven by symbolic world completion before plan correction.
>
> **W3: Abstraction in symbolic memory**
>
> **A3**: We thank the reviewer for this constructive suggestion. We additionally evaluate an **AbstractMem variant**, which uses the VLM to distill reusable abstract rules from feedback. These results suggest that abstract memories can provide reusable high-level guidance, but their benefit is limited in cases where the VLM must reason about scene-specific action semantics or perceive environment-specific details. In these situations, precise predicate-level grounding remains more effective. We will add the results and clarify this motivation in the final version.
>
> |Memory Variant|SR|#PlanRefine|#EnvEvolve|
> |-|-|-|-|
> |RawMem|70.32|5.84|3.63|
> |AbstractMem|80.26|3.08|1.55|
> |SASMem|81.55|2.15|0.84|

---

> > ### Author Rebuttal · Reviewer_p4ht · 2026-04-02
> >
> > I appreciate the authors’ careful and detailed response. The concerns I raised regarding the weakness have been addressed.
> > However, I remain cautious about such a core issue being resolved within such a short period of time.
> >
> > Therefore, I will keep my original score.

---

### Decision · Program_Chairs · 2026-04-30

**Decision:**

Accept (regular)

**Comment:**

The submission introduces a self-adaptive symbolic planning framework that supports refining action plans and evolving the symbolic world.  Reviewers liked the idea but were concerned about its applicability to the open-ended real world.  The authors did a good job during the rebuttal, after which all reviewers turned positive, despite remaining concerns about its presentation and practical usefulness.  The AC would like to follow the reviewers and recommend weak accept. The authors are encouraged to review the submission for the camera-ready.